# Oxide-hybridized carbon as a catalyst support for efficient anion exchange membrane water electrolysis

Jong Seok Park[1,2,11], Hyung-Kyu Lim [3,11], Chuan Hu [4,5,11], Eungjun Lee[1], Hyo Sang Jeon [6], Jongkyung Ryu[7], Sion Oh[1,8], Tae Kyung Lee[1,2], Subin Park[1], Hyeon Keun Cho[5], Seung-Ho Yu [2], Docheon Ahn [9], Young Moo Lee [5] ✉, Myeong-Geun Kim [1,10] ✉ & Sung Jong Yoo [1,10] ✉

The performance of anion exchange membrane water electrolysis, a key technology for achieving net-zero carbon emissions, can be improved by introducing an appropriate catalyst support. However, the ideal support material for water electrolysis remains debatable. Recent efforts focused on enhancing the corrosion resistance by using highly crystalline carbon as a carbon-based support. Despite this progress, the defects intrinsic to carbon emphasize the need for effective passivation strategies to ensure long-term stability and reliability. Addressing this challenge, here we introduce Ti to passivate these surface defects, resulting in oxide-hybridized supports with significantly improved corrosion resistance. The layered double hydroxide catalyst loaded on the Ti-hybridized carbon demonstrates notable performance (8.5 A cm$^{-2}$ at 2 V) and durability (0.17 mV h$^{-1}$ over 900 h at 1 A cm$^{-2}$). The enhanced activity can be attributed to the efficient OH$^-$ supply, as confirmed by *in*-situ Fourier-transform infrared spectroscopy measurements and theoretical calculations. This study provides a foundation for the development of advanced catalyst supports for water electrolysis.

Hydrogen has emerged as a potential energy source and carrier due to its high energy density (33.3 kWh kg$^{-1}$), low weight, and environmental friendliness[1]. Depending on the production method, hydrogen is categorized as gray, blue, green, turquoise, or pink[2]. Among these, green hydrogen is the most desirable because it is produced by an emission-free process, namely water electrolysis (WE). The three main WE technologies are alkaline water electrolysis (AWE), proton exchange membrane water electrolysis (PEMWE), and anion exchange membrane water electrolysis (AEMWE). AWE is mature and well-established, but the diaphragm presents challenges such as gas crossover, mechanical degradation, and stability issues[3]. Consequently, membrane-based systems have gained attention, with PEMWE commercialized since the 1960s. However, the widespread construction of PEMWE facilities has been limited by their high CAPEX,

[1]Center for Hydrogen and Fuel Cells, Korea Institute of Science and Technology (KIST), Seoul, Republic of Korea. [2]Department of Chemical and Biological Engineering, Korea University, Seoul, Republic of Korea. [3]Department of Chemical Engineering , Kangwon National University, Chuncheon, Gangwon-do, Republic of Korea. [4]School of Energy and Environment, Southeast University, Nanjing, Jiangsu Province, China. [5]Department of Energy Engineering, Hanyang University, Seoul, Republic of Korea. [6]Sustainable Energy Research Division, Korea Institute of Science and Technology (KIST), Seoul, Republic of Korea. [7]Department of Materials Science and Engineering, Pohang University of Science and Technology (POSTECH), Pohang, Gyeongbuk, Republic of Korea. [8]Department of Chemical Engineering, Kyung Hee University, Yongin-si, Gyeonggi-do, Republic of Korea. [9]PLS-II Beamline Department, Pohang Accelerator Laboratory, Pohang, Republic of Korea. [10]Division of Energy & Environment Technology, KIST School, University of Science and Technology (UST), Seoul, Republic of Korea. [11]These authors contributed equally: Jong Seok Park, Hyung-Kyu Lim, Chuan Hu. ✉e-mail: ymlee@hanyang.ac.kr; mgkim@kist.re.kr; ysj@kist.re.kr

primarily driven by the high cost of Ir catalysts, which are essential for the oxygen evolution reaction (OER)[4]. In next-generation water electrolyzers (AEMWE), Ir catalysts may be replaced with non-precious metal catalysts, such as transition metal oxides/hydroxides/oxyhydroxides (e.g., FeNiOOH, perovskite oxide, and rutile oxide). However, AEMWE performs unsatisfactorily compared to PEMWE, owing to the lower intrinsic activity and stability of non-precious metals compared to Ir. This performance gap is exacerbated by the inefficient catalyst layer (CL), a problem also observed for PEMWE.

The high anodic potential causes the degradation and corrosion of support materials, dense anode CLs without supports are typically used. To achieve performance comparable to PEMWE, transition metal-based catalysts require the development of efficient CLs. One approach involves directly coating catalysts onto porous transport layers (e.g., Ni foam) to enhance mass transport and bubble release[5]. However, porous transport electrodes present challenges, including inefficient binder utilization and difficulties in optimizing CL structure[6]. Therefore, incorporating catalyst-support materials is essential for the fabrication of efficient CLs. Many researchers have explored oxides (e.g., $TiO_x$, $WO_x$, and $SnO_x$) as support materials for OER catalysts[7–9], although no clear consensus has been reached.

Carbon supports offer an alternative to oxides despite their previous exclusion because of their susceptibility to corrosion under OER conditions. Unlike metal oxides, carbon materials have high electrical conductivities and low packing densities that enable efficient CL fabrication. Recent studies have shown that carbon corrosion can be mitigated by minimizing interactions with water[10]. Hydrophobic and highly crystalline carbon (HCC) supports have extended the AEMWE operation to ~1000 h, garnering significant interest. Nevertheless, this durability remains insufficient for long-term commercialization (100,000 h)[11]. This may be attributable to the intrinsic defects in the HCC, which would need to be passivated to improve the long-term performance.

In this study, the surface defects of HCC were successfully passivated by Ti-O-C species to enhance the durability of the catalyst support. While various oxides (i.e., Ta, Sc, and Zr) can serve a similar role, Ti was selected for its oxophilic properties, promoting efficient $OH^-$ transport during water electrolysis. Specifically, the Ti species were coordinated at the in-plane defect sites of the HCC, as evidenced by Raman spectroscopy, FT-IR spectroscopy, and density functional theory (DFT) calculations. A representative OER catalyst, FeNi-LDH, was loaded onto the Ti/HCC support, exhibiting enhanced durability both in half-cell (1800 h at 0.1 A cm$^{-2}$) and single cell (870 h at 1.0 A cm$^{-2}$) tests. Moreover, the catalyst enhanced the AEMWE performance (current density of 8.5 A cm$^{-2}$ at 2.0 V). DFT calculations and in-situ FT-IR analyses suggested that the enhanced performance may be due to the oxophilic characteristics of Ti, which attracts $H_2O$ and reactant $OH^-$ ions. This enrichment of reactants near the catalyst surface enhance reaction efficiency, thereby improving OER selectivity. Additionally, the improved OER selectivity enables its use in seawater electrolysis. Our results demonstrate that hybridizing metal oxides with carbon not only improves the AEMWE performance but also increase reactant availability, highlighting the importance of developing hybrid catalyst supports. We propose a new strategy to suppress carbon corrosion, and expect this study to serve as a foundation for developing ideal support materials comprising a higher proportion of oxide and a lower proportion of carbon without compromising the performance.

## Results

### Passivation of carbon defects via hybridization with Ti

Although carbon-based supports have not been widely used in water electrolysis owing to carbon corrosion, our group recently demonstrated corrosion-resistant HCC[10]. Unfortunately, the inevitable intrinsic defects of HCC necessitate passivation for long-term performance. Here, Ti, in the form of titanium (IV) isopropoxide, was introduced to passivate the surface defects of the HCC, and resulted in the formation of a distinctive covalent Ti-O-C bond (Fig. 1a)[12,13]. Three Ti-hybridized HCC ($Ti_{xwt\%}$/HCC) samples with varying Ti contents (x = 4, 9, and 13) were prepared by using different amounts of the Ti precursor and using a sonochemically assisted method. The Ti content was verified using inductively coupled plasma optical emission spectrometry (ICP-OES). The synthesis procedure is described in the Methods section.

X-ray photoelectron spectroscopy (XPS) and Fourier-transform infrared spectroscopy (FT-IR) measurements clearly indicate that Ti-O-C bonds were generated on the HCC surface. The XPS Ti 2$p$ analysis revealed that, in $Ti_{xwt\%}$/HCC, Ti has two oxidation states (corresponding to $Ti^{4+}$ and the Ti-O-C bonds) (Fig. 1b). To quantitatively compare the Ti-O-C content in the $Ti_{xwt\%}$/HCC samples, the Ti content determined by ICP-OES was calibrated as a ratio of the Ti 2$p$ area to that of C 1$s$. For the $Ti_{4wt\%}$/HCC, $Ti_{9wt\%}$/HCC, and $Ti_{13wt\%}$/HCC samples, the sample with the highest Ti content exhibited a greater number of Ti-O-C bonds, corresponding to 0.83, 1.70, and 2.80 wt.%, respectively. A similar trend was observed for C 1$s$, where the atomic composition of Ti-O-C increased to 12, 23, and 33 at.%, respectively. Additionally, the shifts of the Ti 2$p$ peak (458.9 eV) of the Ti-O-C bond and the C 1$s$ peak to lower and higher binding energies, respectively, are indicative of electronic interactions between Ti and C (Supplementary Fig. S1). FT-IR analysis consistently pointed to Ti-O-C bond formation in $Ti_{xwt\%}$/HCC (Fig. 1c)[14]. Additionally, the electrical conductivity of $Ti_{4wt\%}$/HCC, $Ti_{9wt\%}$/HCC, and $Ti_{13wt\%}$/HCC was comparable (Supplementary Fig. S2). Transmission electron microscopy (TEM) images and extended X-ray absorption fine structure (EXAFS) revealed that most Ti exists as Ti-O-C and in clusters; a small amount of $TiO_2$ is detected only when excess Ti precursor (above 0.5 mmol) is used (Supplementary Figs. S3, S4). XRD peaks corresponding to $TiO_2$ were not observed (Supplementary Fig. S5).

Raman spectroscopy analyses and DFT calculations indicated that the Ti-O-C bond was preferentially formed at the defect sites within the HCC. HCC exhibited three Raman peaks, G, $D_1$, and $D_2$, which corresponded to graphitic carbon, edge defects, and planar defects, respectively (Supplementary Fig. S6). The $I_{D1}/I_G$ and $I_{D2}/I_G$ ratios were determined to quantify the concentration of each defect type. The $I_{D2}/G$ ratio decreased as the Ti content increased (Fig. 1d), suggesting that Ti tended to passivate the planar defects of the HCC through the Ti-O-C bond. The $I_{D1}/I_G$ ratio of $Ti_{xwt\%}$/HCC exceeded that of pristine HCC, although this ratio decreased with increasing Ti content. These results suggest that the passivation effect on the in-plane defects may be greater than that on the edge defects when Ti is incorporated into HCC[15]. In Supplementary Fig. S7, HR-STEM images show that most Ti single atoms (marked with yellow circles) in $Ti_{9wt\%}$/HCC are located at in-plane defect sites of the HCC.

The role of hybridization in improving corrosion resistance is further supported by DFT calculations, which compare the interactions of the Ti-O moiety with various defect sites in HCC. The binding energy (BE) of the Ti-O moiety at different defect configurations (i.e., $V_1$, $V_{2-1}$, $V_{2-2}$, and $V_3$) was calculated, where $V_1$ and $V_3$ represent radical defect sites, and $V_{2-1}$ and $V_{2-2}$ represent $sp^2$ conjugated topological defect sites (Supplementary Figs. S8, S9 and Supplementary Data 1). Perfect graphene (Gr) showed no binding affinity toward the Ti-O moiety due to its complete $sp^2$ conjugation. Notably, the $V_1$ and $V_3$ sites exhibited strong BEs of −4.23 and −4.49 eV, respectively, which are energetically favorable, comparable to the cohesive energy of bulk $TiO_2$ (−5.86 eV). This suggests that the Ti-O moiety can passivate unstable radical carbon sites, thereby contributing to surface stabilization. In contrast, the BEs of $V_{2-1}$ and $V_{2-2}$ were relatively weak (−0.95 and −1.45 eV, respectively), due to the energy cost associated with breaking the $sp^2$ conjugation network. The passivation effect was further demonstrated by the electrochemical (EC) oxidation behavior of the carbon sites. As

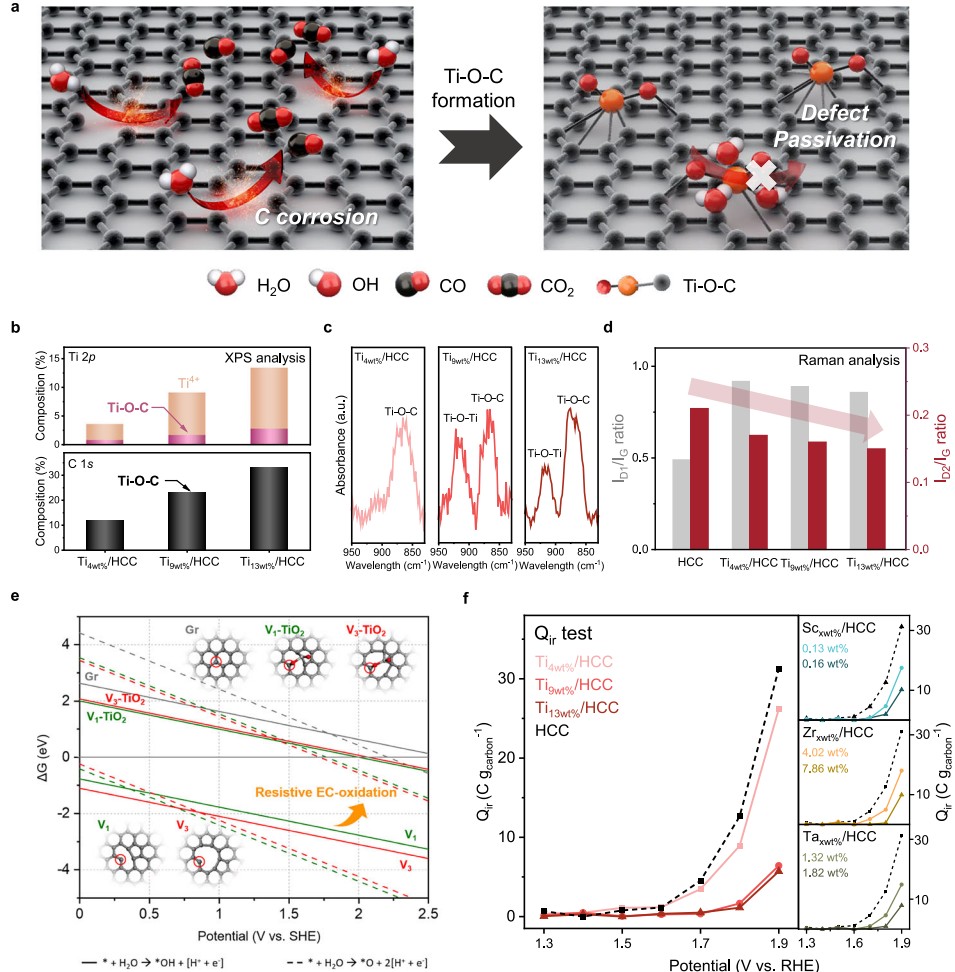

**Fig. 1 | Fabrication and characterizations of Ti-hybridized HCC. a** Schematic illustration of the corrosion resistance mechanism of oxide-hybridized carbon by passivating intrinsic defects in HCC. **b** XPS analysis of Ti 2$p$ and C 1$s$ for Ti$_{xwt\%}$/HCC (x = 4, 9, and 13). **c** FT-IR spectra and **d** Raman analysis (left axis: $I_{D1}/G$ ratio; right axis: $I_{D2}/G$ ratio) results. **e** Gibbs free energy (ΔG) diagrams for electrochemical oxidation on radical carbon sites with and without TiO$_2$ passivation. **f** Irreversible charge (Q$_{ir}$) test result for M$_{xwt\%}$/HCC (M = Ti, Sc, Zr, and Ta). Source data are provided as a Source Data file.

shown in Fig. 1e (Supplementary Data 1), pristine radical sites (V$_1$ and V$_3$) exhibit highly negative ΔG values for both *OH and *O formation, indicating their vulnerability to oxidative corrosion over a wide potential range of 0−2.5 V$_{SHE}$. The sequential formation of *OH (solid lines) and *O (dashed lines) species proceeds spontaneously at these radical sites, thereby initiating carbon corrosion. In contrast, when these sites are passivated by TiO$_2$ (designated as V$_1$-TiO$_2$ and V$_3$-TiO$_2$), they are remarkably resistant to oxidation, with positive ΔG values maintained up to -1.8 V$_{SHE}$. This shift in oxidation resistance demonstrates that TiO$_2$ passivation effectively stabilizes the reactive radical sites, which renders them resistant to corrosion, comparable to perfect Gr. The substantial enhancement in the oxidation resistance, particularly in the operating potential range of water electrolysis, suggests that TiO$_2$-passivated carbon is appropriate to serve as a stable support material under harsh oxidative conditions.

Using the resulting Ti-hybridized HCC, we verified the enhanced carbon corrosion resistance by conducting a Q$_{ir}$ (irreversible charge) test to quantify the degree of irreversible carbon oxidation (Fig. 1f)[16]. Compared with pristine HCC, the corrosion resistance was significantly enhanced as the Ti content increased. The enhancement in corrosion resistance by Ti decoration was more pronounced when low-crystalline carbon was used owing to the abundance of defect sites (Supplementary Fig. S10). Moreover, a similar trend was observed when M-O-C was varied (M = Ta, Sc, and Zr). This result confirms that

the hybridization of HCC and oxide materials is an effective strategy for improving the corrosion resistance of carbon.

## Synthesis and characterization of Ti/HCC-supported LDH catalysts

We used corrosion-resistant Ti$_{xwt\%}$/HCC as a supporting material for the OER catalyst, a layered double hydroxide (LDH), one of the most representative non-precious metal catalysts for AEMWE. The synthesis, a one-pot sonochemical-assisted method, consisted of five stages: (I) adding the Ti precursor to the carbon-dispersed ethylene glycol (EG) solution, (II) adding the Fe and Ni precursors (dispersed in deionized (DI) water) to the above solution, (III) gradually adding 66 mg of the reducing agent (NaBH$_4$) from a total of 132 mg, (IV) adding the remaining 66 mg of NaBH$_4$, and (V) allowing the reaction to proceed for 60 min. The same procedure was followed for the preparation of Ti$_{xwt\%}$/HCC, except for the introduction of the metal precursors. Details appear in the Methods section. Figure 2a shows the *ex*-situ EDS mapping images at each stage of the process (Supplementary Fig. S11). In Stage I, Ti was uniformly distributed on the HCC surface. Interestingly, when the Fe and Ni precursors were added in Stage II, EDS mapping did not reveal any Fe, Ni, or Ti on the surface. After the addition of NaBH$_4$ in Stage III, EDS showed that Fe and Ti mostly resided on the surface of the HCC, whereas Ni subsequently made its appearance on the outer surface in Stage IV. As the reaction proceeded

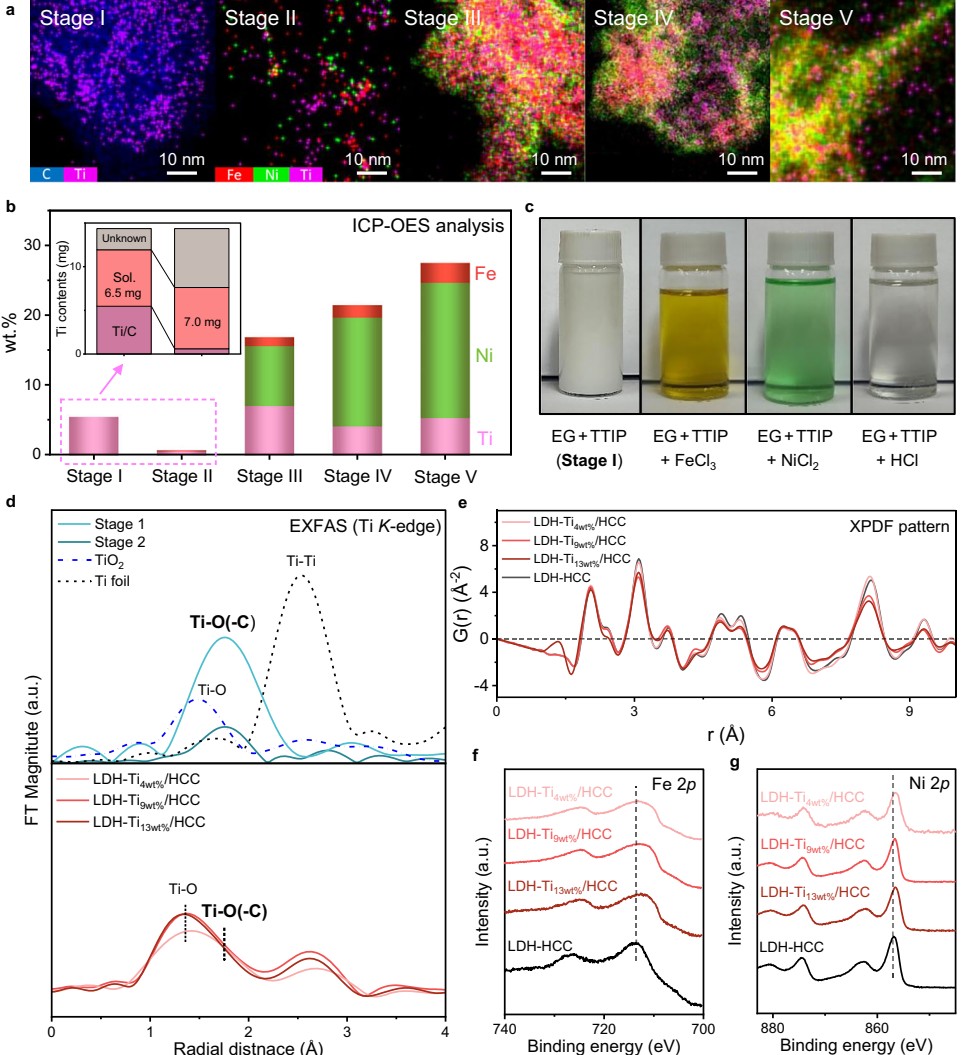

**Fig. 2 | Synthesis process and material characterizations of LDH-Ti$_{xwt\%}$/HCC.**
**a** EDS mapping images of the sample obtained at each reaction stage for LDH-Ti$_{9wt\%}$/HCC. Metal content at each stage, determined by **b** ex-situ ICP-OES. **c** Changes in the transparency of EG + TTIP solution due to chloride ion injection. **d** Extended X-ray absorption fine structure spectra (EXFAS) for the Ti $K$-edge of (up) stage I and stage II, and (down) LDH-Ti$_{xwt\%}$/HCC. **e** X-ray atomic pair distribution function (XPDF) and XPS results **f** Fe 2$p$ and **g** Ni 2$p$ for LDH-Ti$_{xwt\%}$/HCC. Source data are provided as a Source Data file.

to Stage V, Fe and Ni engaged in alloying, resulting in a well-mixed LDH structure.

The aforementioned observations were subsequently investigated in greater detail. ICP-OES and XPS analysis of the powder and supernatant solutions at Stages I and II informed that the Ti content on the carbon support decreased dramatically upon dispersion of the metal precursor (FeCl$_3$ and NiCl$_2$) solutions into the Stage I solution (Fig. 2b), whereas the Ti content of the solution (inset of Fig. 2b) increased. This led us to propose that the introduction of Cl$^-$-containing metal precursors (e.g., FeCl$_3$ and NiCl$_2$) into the Stage I solution results in the formation of Ti(OH)$_4$—widely accepted to form when titanium (IV) isopropoxide is added to EG[17]—after which it is etched into small cluster complexes[18]. This hypothesis was confirmed by injecting metal chloride solutions (dispersed in EG) or HCl solution into the opaque TTIP/EG solution, which became transparent (Fig. 2c). In contrast, metal precursors without chloride ions did not affect the opacity, indicating that the Cl$^-$ ions were acting as etchants (Supplementary Fig. S12). The effect of DI water was excluded because the transparency was not observed to change when DI was injected (Supplementary Fig. S13).

EXAFS spectra indicated the formation of Ti-O-(C) bonds in the HCC, followed by the subsequent formation of the LDH on the Ti-O-C sites (Fig. 2d). Both the Ti $K$-edge spectra acquired during Stage I and II clearly indicate the formation of Ti-O-(C) bonds (-1.88 Å), as inferred from the bond lengths of Ti-O (1.88-1.95 Å) and Ti-C (2.06-2.25 Å) in the optimized DFT model. LDH-Ti$_{xwt\%}$/HCC gives rise to a broad peak representing the Ti-O (1.36 Å) and Ti-O(-C) (1.88 Å) bonds. The broadening of the Ti-O(-C) peak compared to that of Ti$_{xwt\%}$/HCC suggests that the LDH catalyst was formed on the Ti-O-C species. The Ti-O peak possibly originates from unwanted anatase TiO$_2$ and/or Ti dopant ions in the LDHs[19]. The X-ray pair distribution function (XPDF), XRD patterns, and EXAFS spectra consistently showed that Ti incorporation caused no significant structural differences, suggesting that the majority of the Ti existed in the form of Ti-O-C rather than as ternary FeNiTi LDHs (Fig. 2e, Supplementary Figs. S14, S15a, b). Strong electronic interaction between Fe/Ni and Ti was evident from the XPS results, indicating that the LDH was anchored at the Ti-O-C sites (Fig. 2f, g, and Supplementary Fig. S16). Morphological and structural characterizations by STEM and EXAFS show that there was no significant structural and morphological difference between LDH-Ti$_{9wt\%}$/HCC and LDH-HCC (Supplementary Fig. S15c, d); only a slight change in oxidation states of Fe and Ni was observed by XPS analyses (Supplementary Figs. S17, S18).

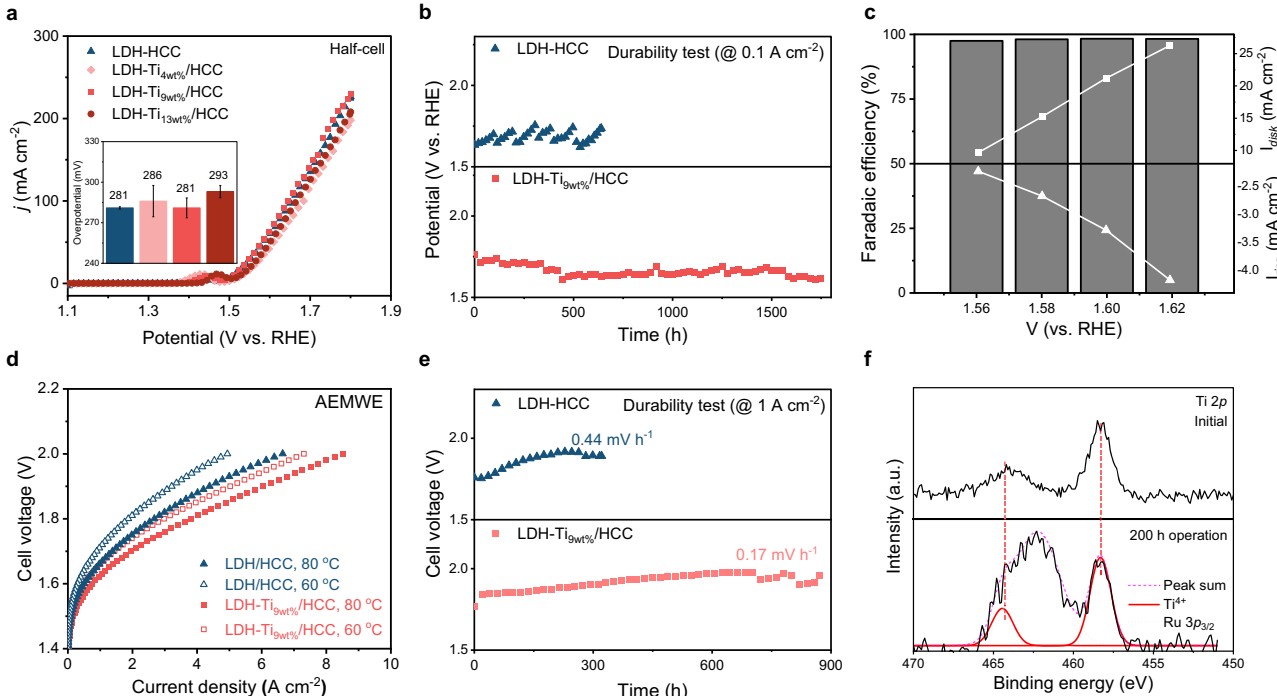

**Fig. 3 | Evaluation of the electrocatalytic OER performance of LDH-Ti/HCC.**
**a** LSV curves for OER of LDH-Ti$_{xwt\%}$/HCC (inset: overpotential at 10 mA cm$^{-2}$).
**b** Long-term durability test via chronopotentiometry conditions for 0.1 A cm$^{-2}$ in a half-cell setup. **c** Faradaic efficiency of OER, along with the I$_{disk}$ and I$_{ring}$ values, measured over a voltage range of 1.56 to 1.62 V$_{RHE}$. **d** Current–Voltage (I–V) curves for AEMWE measured at 60 °C and 80 °C. **e** AEMWE durability test at 1 A cm$^{-2}$. **f** XPS

Ti 2$p$ measured before and after 200 h of AEMWE operation. Half-cell test were conducted at room temperature (25 °C) in 1 M KOH (pH 14.0) using an RDE (32 μg cm$^{-2}$, 2400 rpm). Durability test were conducted on a SUS mesh with a catalyst loading of ~1.6 mg cm$^{-2}$. The measured potentials were not IR corrected (R$_s$ = 4.3 ± 0.11 Ω). Source data are provided as a Source Data file.

## Evaluation of electrocatalytic OER performance of LDH-Ti/HCC

The LDH-Ti$_{xwt\%}$/HCC catalyst notably enhanced the AEMWE performance, especially as a result of the improved catalyst durability. The OER catalytic activity and long-term durability of LDH-Ti$_{xwt\%}$/HCC were evaluated in both half-cell and AEMWE cell configurations. The OER overpotentials for 10 mA cm$^{-2}$ for LDH-HCC, LDH-Ti$_{4wt\%}$/HCC, LDH-Ti$_{9wt\%}$/HCC, and LDH-Ti$_{13wt\%}$/HCC were 282, 302, 282, and 300 mV, respectively (Fig. 3a). Thus, the OER activity of LDH-Ti$_{9wt\%}$/HCC was slightly higher than others, making it selected for further measurements. In the absence of Ti/HCC support, aggregated LDH was formed (Supplementary Fig. S19), which shows inferior OER activity compared to LDH-Ti$_{9wt\%}$/HCC (Supplementary Fig. S20). Additionally, the current density was normalized by actual metal loading amount (Supplementary Fig. S21). The higher Ti loading led to lower OER activity, suggesting that Ti does not contribute to OER activity. Figure 3b presents the results of the catalytic durability test under chronopotentiometric conditions (at 0.1 A cm$^{-2}$). The LDH-Ti$_{9wt\%}$/HCC demonstrated stable durability for ~1800 h, whereas the LDH-HCC maintained stability for only 350 h. This result clearly indicates that the formation of Ti-O-C bonds enhanced the corrosion resistance and structural stability of the LDH (Supplementary Fig. S22), attributed to the anchoring of LDH at the Ti-O-C sites (Supplementary Fig. S23). Additionally, these experiments were conducted using a small amount of metal (170 μg$_{Fe,Ni}$ cm$^{-2}$), due to the enhanced catalyst utilization by carbon supports. The Faradaic efficiency (FE) of the OER was determined using a rotating ring disk electrode (RRDE) test (Fig. 3c). The FE of the LDH-Ti$_{9wt\%}$/HCC was approximately 100% above 1.56 V$_{RHE}$[20]. In addition, Tafel slopes, turnover frequency (TOF) values, and Nyquist plots were obtained (Supplementary Fig. S24). There was only a slight change in R$_{ct}$ upon Ti introduction, implying that the electrical conductivity was not compromised.

The AEMWE performance of LDH-Ti$_{9wt\%}$/HCC is comparable to that of state-of-the-art electrocatalysts[10,21–27]. Unlike the slight difference in activity observed for the half-cell configuration due to the abundance of reactants, the effect of Ti on the AEMWE performance was notable. LDH-HCC and LDH-Ti$_{9wt\%}$/HCC delivered current densities of 8.51 A cm$^{-2}$ and 6.65 A cm$^{-2}$ at 2.0 V, respectively (Fig. 3d). A clear discrepancy in the cell voltage was observed in the high current density region, which could be attributed to the difference in mass transport efficiency[28]. The enhanced mass transport efficiency of LDH-Ti$_{9wt\%}$/HCC is discussed in the next section. Apart from mass transport, the effect of carbon corrosion on the AEMWE performance was investigated by using SEM, XPS, and UV-vis absorption spectroscopy. Although no noticeable change in catalyst layer thickness was observed during the 200 h of AEMWE test (Supplementary Fig. S25a, b), XPS C 1s spectra revealed a stronger carbonate peak in LDH/HCC, compared to LDH-Ti/HCC (Supplementary Fig. S25c). Meanwhile, the extent of metal dissolution varied significantly depending on the presence of Ti. The corrosion-resistant Ti/HCC effectively suppressed metal dissolution during electrolysis (Supplementary Figs. S25d, e, S26)[29]. The difference likely originated from corrosion resistance, as evidenced by the changes in the I$_{D1}$/I$_G$ ratio (Supplementary Fig. S27). AEMWE durability evaluations are typically conducted at low current densities (<0.5 A cm$^{-2}$)[30,31]. In contrast, we employed a harsher condition of 1.0 A cm$^{-2}$ for the chronopotentiometry test. LDH-Ti$_{9wt\%}$/HCC demonstrated comparable durability by maintaining its stability for ~1000 h with a degradation rate of 0.17 mV h$^{-1}$, whereas LDH-HCC deteriorated for 450 h of reaction at 0.44 mV h$^{-1}$ (Fig. 3e). This result clearly indicates that the carbon support is effective and that the incorporation of Ti significantly enhances the catalyst durability. This is attributable to the high OER selectivity, which suppresses unwanted corrosion reactions, and the anchoring of LDHs, which helps prevent

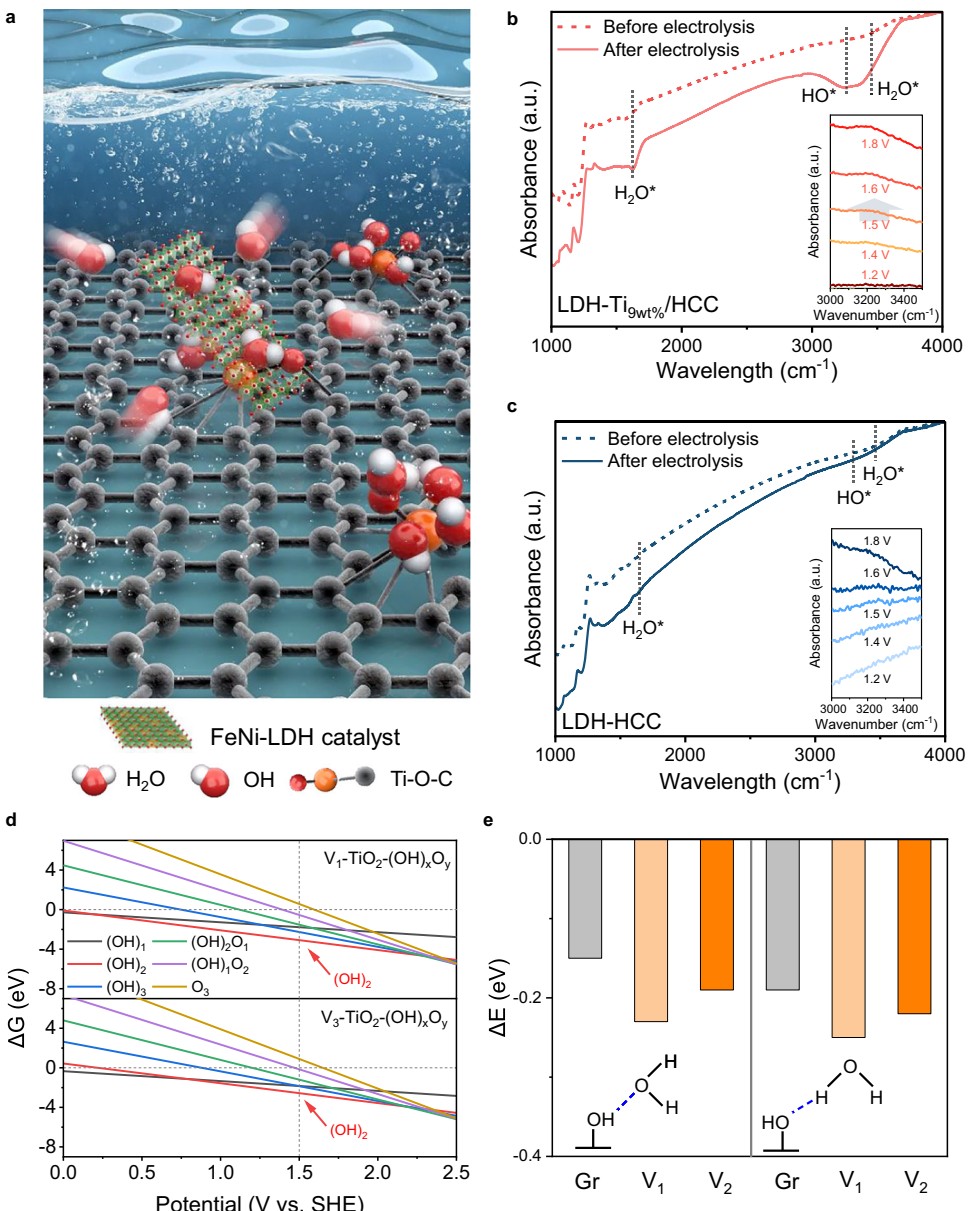

**Fig. 4 | Elucidating the role of Ti in water electrolysis. a** Schematic illustration of the attraction of OH⁻ from the solution to LDH by the HCC surface decorated with Ti. FT-IR spectra of **b** LDH-Ti$_{9wt\%}$/HCC and **c** LDH/HCC, obtained before (dotted) and after (solid) electrolysis in 1 M PBS (inset: in-situ FT-IR measured from OCV to 1.8 V in 1 M PBS). **d** Free energy diagrams showing stepwise oxidation pathways of Ti sites at $V_1$ and $V_3$ defects. **e** Water binding energies at oxygen-functionalized graphene (Gr) and Ti-passivated sites ($V_1$/$V_3$-TiO$_2$-(OH)$_2$). Source data are provided as a Source Data file.

catalyst degradation. Additionally, Ti $2p$ XPS spectra confirmed that the chemical state of Ti remained largely unchanged after the durability test. (Fig. 3f). The durability of various recently developed OER catalysts is compared in Supplementary Fig. S28 and Supplementary Table S1, where the degradation rate is shown as a function of the plotted time[10,21–27].

Using *ex*-situ and *in*-situ FT-IR analysis and DFT calculations, we suggest that the enhanced AEMWE performance is attributed to the ability of Ti to attract H$_2$O molecules and OH⁻ ions. Oxophilic Ti attracts H$_2$O and OH⁻ from the electrolyte, which increases the concentration of OH⁻ ions near the LDH catalysts, as illustrated in Fig. 4a[32]. The strong tendency of OH⁻ ions and H$_2$O to move toward Ti was confirmed by ex-situ and in-situ FT-IR analyses. The LDH-Ti$_{9wt\%}$/HCC sample exhibited distinct peaks corresponding to H$_2$O* (~1600 cm⁻¹ and ~3300 cm⁻¹) and OH* (~3100 cm⁻¹) after electrolysis (Fig. 4b)[33]. This result was further confirmed by *in*-situ FT-IR measurements, wherein

the peak at ~3100 cm⁻¹ appeared above 1.5 V$_{RHE}$ (inset of Fig. 4b and Supplementary Fig. S29). In contrast, the H$_2$O* and OH* peaks were barely observed for LDH-HCC (Fig. 4c). The experimental details are presented in the SI. The peak at ~3250 cm⁻¹ can be deconvoluted to tetrahedral coordination, while the peak at ~3400 cm⁻¹ is attributed to trihedral coordination (Supplementary Fig. S30). A higher proportion of tetrahedrally coordinated water was observed on LDH–Ti/HCC. However, the detailed correlation between interfacial water coordination and electrochemical performance is beyond the scope of this study. These results suggest that Ti species on HCC play a key role in attracting OH⁻ ions, thereby contributing to the improved single-cell performance. Additionally, the LDH supported on TiO$_2$/HCC and the LDH/HCC physically mixed with TiO$_2$ exhibited comparable activity to LDH-Ti$_{9wt\%}$/HCC, further confirming the role of Ti. In contrast, the LDH-TiO$_2$ mixed with HCC showed poor activity, indicating that a thin Ti layer is essential to ensure efficient electron transfer

between LDH and HCC (Supplementary Figs. S31, S32). To further confirm that the Ti species remain unchanged during the OER process, quasi in-situ XAFS analysis was performed. As shown in Supplementary Fig. S33, the spectra remained unchanged during electrolysis. An additional experiment was conducted to confirm the role of uncovered Ti-O-C sites in efficiently supplying reactants, by adding $F^-$ ions to 1 M KOH, which are capable of poisoning Ti (Supplementary Fig. S34)[34]. With increasing KF concentration, the OER activity of LDH-Ti$_{9wt\%}$/HCC was significantly degraded, especially in the high current density region, whereas that of LDH-HCC remained largely unchanged. DFT calculations further confirmed the unique water-binding characteristics of the Ti-passivated defect sites. The $V_1$-TiO$_2$ and $V_3$-TiO$_2$ models had similar oxidative water adsorption behaviors and were capable of accommodating multiple *OH and *O groups through sequential $H_2O$ oxidation (Fig. 4d and Supplementary Data 1). Thus, the exposed Ti-passivated sites readily formed various *Ti(OH)$_x$O$_y$ configurations under electrochemical oxidation conditions, demonstrating their oxophilic nature. In addition, the *OH- and *O-decorated Ti sites exhibited stronger $H_2O$ binding energies than the oxygen functional groups on Gr. $V_1/V_3$-TiO$_2$-(OH)$_2$ demonstrated enhanced water-binding affinity, regardless of the hydrogen-bonding configuration (Fig. 4e and Supplementary Data 1). This can be attributed to the highly polarized Ti-O bonds arising from the high oxidation state of Ti, which created more negatively charged O sites for effective $H_2O$ binding. This theoretical finding supports our experimental observations that the local water concentration increased near the Ti sites, suggesting their role in facilitating a steady water supply to the catalyst surface during the OER.

### Seawater electrolysis performance of LDH-Ti/HCC

Recently, water electrolysis was extended to the electrolysis of seawater, which constitutes ~96.5% of Earth's water resources, and reduces the reliance on freshwater and the need for purification facilities[35,36]. However, in seawater, the OER competes with the chlorine evolution reaction (CER) because of the presence of excess $Cl^-$ ions[35,37]. Because the catalyst surface is susceptible to poisoning and corrosion by $Cl^-$ ions, enhancing the selectivity of the OER over the CER is essential. Based on the high OER selectivity confirmed by the RRDE test, the seawater-splitting activities were evaluated in both 1 M KOH + 0.5 M NaCl and 1 M KOH + seawater electrolytes. Although there was slight decrease in activity with increasing NaCl concentration of 1 M KOH + 0.5 M NaCl, the activity degradation was not influenced by the presence of Ti (Fig. 5a); likewise, seawater-splitting activity was independent of Ti content. This could be due to the abundance of reactants and similar electronic and structural properties of LDH-Ti$_{9wt\%}$/HCC and LDH-HCC (Supplementary Fig. S35).

Interestingly, the durability of LDH-Ti$_{9wt\%}$/HCC (0.03 mV h$^{-1}$) for 200 h is comparable to that of LDH-HCC (0.70 mV h$^{-1}$) for 100 h (Fig. 5b). A durability test was conducted using chronopotentiometry at 0.1 A cm$^{-2}$. When measuring the OER current in seawater, the current may include contributions from the CER and corrosion reactions. UV-vis spectroscopy was used to confirm that the current mainly originated from the OER. The $OCl^-$ ions generated as a result of the CER were quantitatively analyzed by constructing a calibration curve using a previously reported method (Supplementary Fig. S36)[38]. The calibration curve was constructed by recording the absorption at 289 nm (corresponding to $OCl^-$) of electrolyte aliquots collected at various reaction times (10, 20, 30, 50, 100, 150, and 200 h)[39]. The low FE (below 3%) for the CER over 200 h of reaction indicated that the introduction of Ti successfully enhanced the OER selectivity (Supplementary Fig. S36). This result suggests that the high OER selectivity of LDH-Ti$_{9wt\%}$/HCC enhances the durability under 1 M KOH + 0.5 M NaCl conditions.

LDH-Ti$_{9wt\%}$/HCC was also tested as an OER catalyst in a 1 M KOH + 0.5 M NaCl electrolyzer (Fig. 5c). The LDH-Ti$_{9wt\%}$/HCC exhibited higher current density of 3.4 A cm$^{-2}$ at 2 V, compared to 1.2 A cm$^{-2}$

for LDH-HCC, likely due to the preferential adsorption of $OH^-$ by oxophilic Ti species. The enhanced durability of LDH-Ti$_{9wt\%}$/HCC to that of LDH-HCC indicates that corrosion by $Cl^-$ is effectively suppressed. The LDH-Ti$_{9wt\%}$/HCC produced 2.5 mV h$^{-1}$ for ~140 h, whereas LDH-HCC underwent degradation at the rate of 5.7 mV h$^{-1}$ for ~45 h (Fig. 5d). The enhanced durability seems due to the enhanced OER selectivity by reducing competitive $Cl^-$ adsorption and the undesirable CER. Additionally, we further evaluated AEMWE performance using 1 M KOH + seawater condition (Fig. 5e, f). LDH-Ti$_{9wt\%}$/HCC exhibited a higher current density of 1.1 A cm$^{-2}$ at 2 V, compared to 0.66 A cm$^{-2}$ for LDH-HCC. It is generally accepted that AEMWE performance is degraded by cationic and anion impurities (e.g., $SO_4^{2-}$, $F^-$, $Mg^{2+}$, $Ca^{2+}$, etc.). LDH-Ti$_{9wt\%}$/HCC maintained stability for about 250 h, whereas LDH-HCC maintained stable performance for 80 h. To clarify the cause, we added $SO_4^{2-}$ ions into 1 M KOH (Supplementary Fig. S38). At low concentrations, the addition of $SO_4^{2-}$ appeared to slightly enhance the performance by facilitating LDH reconstruction[40]. However, as the concentration of $SO_4^{2-}$ increases, a significant drop in performance was observed, particularly for LDH-HCC, even at lower $SO_4^{2-}$ concentrations compared to LDH-Ti$_{9wt\%}$/HCC. This is attributed to the blocking of Ni sites, as evidenced by the decreased intensity of Ni redox peaks. These results indicate that Ni sites in LDH-HCC are more susceptible to $SO_4^{2-}$-induced poisoning and suggest that developing electrocatalysts resistant to anionic impurity-induced corrosion is essential for ensuring long-term durability in seawater electrolysis, although the exact mechanism by which Ti mitigates $SO_4^{2-}$ poisoning remains unclear.

## Discussion

In this study, we successfully demonstrated that incorporating Ti-O moieties passivates the in-plane defect sites of HCC, enhancing its corrosion resistance. Among the various oxides studied, Ti was selected because of its oxophilic properties, which enable an efficient supply of the $OH^-$ reactant during water electrolysis. This indicates that oxide decoration not only increases the corrosion resistance but also facilitates the OER. The LDH-Ti/HCC catalyst delivered notable performance with respect to the electrolysis of both pure alkaline water and seawater, attributed to its enhanced OER selectivity. Furthermore, the LDH-Ti/HCC demonstrated durability, surpassing previously reported results and underscoring the potential of carbon-containing supports. The use of oxide-hybridized carbon provides a promising pathway toward the adoption of oxide-based supports, which are not yet widely utilized, and offers valuable insights into the development of ideal OER catalyst supports.

## Methods

### Materials used for M$_{xwt\%}$/HCC (M = Ti, Sc, Ta, and Zr; x = 4, 9, and 13) synthesis

Iron (III) chloride hexahydrate (FeCl$_3$·6H$_2$O, 99%), nickel (II) chloride hexahydrate (NiCl$_2$· 6H$_2$O, 99%), sodium borohydride (NaBH$_4$, 99%), tantalum (V) isopropoxide (C$_{15}$H$_{35}$O$_5$Ta, 99.9%), and ethylene glycol (EG, 99.8%) were purchased from Alfa Aesar (USA). Titanium (IV) isopropoxide [Ti(OCH(CH$_3$)$_2$)$_4$, 97% metal basis], scandium (III) isopropoxide [Sc(OCH(CH$_3$)$_2$)$_3$, 22.0%], and zirconium isopropoxide isopropanol complex [Zr(OCH(CH$_3$)$_2$)$_4$·((CH$_3$)$_2$CHOH), 99.9%] were purchased from Sigma Aldrich (USA). All chemicals were used as received without purification. High crystalline carbon (HCC) was prepared by annealing Ketjen black (KB-600JD) at 2700 °C under argon for 2 h.

### Synthetic procedure for Ti$_{xwt\%}$/HCC, LDH-Ti$_{xwt\%}$/HCC, FeNi LDH, and LDH-TiO$_2$/HCC

Ti$_{xwt\%}$/HCC was prepared by adding 100 mg of HCC to 80 mL of EG, and the solution was homogenized by ultrasonication (40 kHz, 300 W, JAC-3010, U1Tech) for 15 min. The Ti content of Ti$_{xwt\%}$/HCC was varied

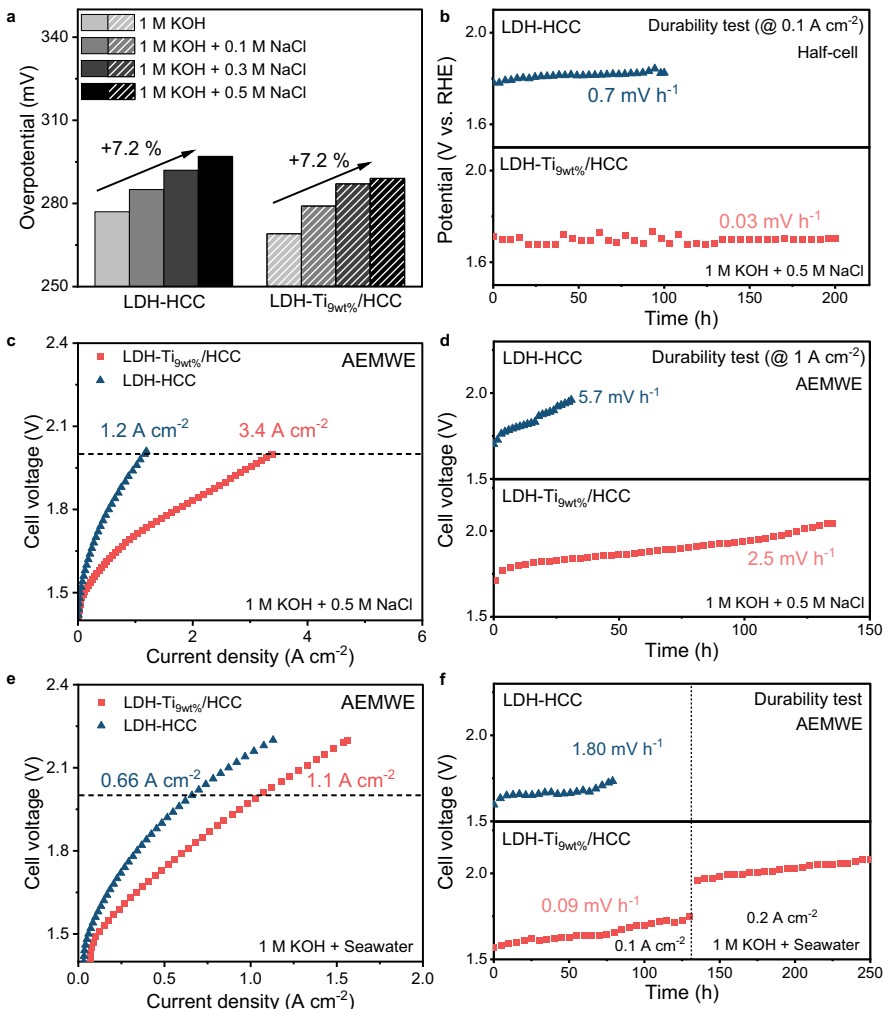

**Fig. 5 | Seawater splitting performance of LDH-Ti/HCC. a** LSV curve for OER of LDH-HCC and LDH-Ti$_{xwt\%}$/HCC (inset: overpotential at 10 mA cm$^{-2}$). **b** Long-term durability test via chronopotentiometry conditions for 0.1 A cm$^{-2}$ in a half-cell setup. **c** I−V curves for AEMWE operated under 1 M KOH + 0.5 M NaCl condition (pH 14).

**d** AEMWE durability test in 1 M KOH + 0.5 M NaCl at 1 A cm$^{-2}$. **e** I-V curves and **f** durability test result for AEMWE operated under 1 M KOH + seawater (pH 13.8) conditions. Source data are provided as a Source Data file.

by adding different amounts (29, 87, and 138 µL) of Ti[OCH(CH$_3$)$_2$]$_4$ solution to the above-mentioned HCC solution to prepare Ti$_{4wt\%}$/HCC, Ti$_{9wt\%}$/HCC, and Ti$_{13wt\%}$/HCC, respectively. Lastly, the reducing agent (NaBH$_4$) dissolved in 10 mL of DI water was slowly dropped into the Ti$_{xwt\%}$/HCC solutions at a rate of 2 mL min$^{-1}$. Different amounts of NaBH$_4$ (116, 132, and 148 mg) were used to fully reduce Ti$_{4wt\%}$/HCC, Ti$_{9wt\%}$/HCC, and Ti$_{13wt\%}$/HCC, respectively. In addition to Ti$_{xwt\%}$/HCC, three different M$_{xwt\%}$/HCC samples (M= Zr, Ta, and Sc) were prepared using the same synthesis process. The LDH-catalyst-coated Ti$_{xwt\%}$/HCC was synthesized via a one-pot process. First, the metal stock solution was prepared by dissolving FeCl$_3$·6H$_2$O (90 mg) and NiCl$_2$·6H$_2$O (273 mg) in 5 mL of DI water, respectively. Next, this stock solution was added to the Ti$_{xwt\%}$/HCC solution under ultrasonication. Subsequently, the reducing agent was prepared, as described above, and added to the solution drop wise, following which the reaction was allowed to proceed for 90 min below 50 °C. The products were collected via vacuum filtration and dried in a vacuum oven for further characterization. For FeNi LDH synthesis, FeCl$_3$·6H$_2$O (90 mg) and NiCl$_2$·6H$_2$O (273 mg) solutions were prepared by dissolving each precursors in 5 mL of DI water, respectively. After the precursor solutions were added to the 80 mL of EG, the suspension was dispersed by bath

sonicator for 30 min. Once the NaBH$_4$ (132 mg) was dissolved in 10 mL of DI water, the solution was added dropwise to the above suspension at a rate of 2 mL min$^{-1}$. All the synthesized samples were collected via vacuum filtration and dried in a vacuum oven for overnight.

## Material characterization

The oxidation state and chemical composition of the electrocatalysts were determined with X-ray photoelectron spectroscopy (XPS, Nexsa, Thermo Fisher Scientific™, USA) using Al Kα (0.83386 nm) as X-ray source. For the structural analyses, X-ray diffraction (XRD, D/MAX 2500PC, Rigaku, Japan), X-ray pair distribution function (XPDF, Empyrean, Malvern Panalytical, UK), and *in*-operando XRD (X'pert pro, Malvern Panalytical, UK) were used. The structure of HCC was analyzed using Raman spectroscopy (532 nm laser, in-Via Raman microscope, Renishaw, UK). Fourier-transform infrared (FT-IR, NiCOLET iS 10, Thermo Fisher Scientific) spectroscopy was conducted to study the chemical compositions and reaction intermediates. Morphological analyses were conducted by using (scanning) transmission electron microscopy [(S)TEM, Titan™ 80-300, FEI, USA] and energy dispersive X-ray spectroscopy (EDS, Talos F200X, Thermo Fisher Scientific™, USA). The amounts of metals (Fe, Ni, Ti, Ta, Sc, and Zr) loaded on the

HCC were determined using inductively coupled plasma optical emission spectroscopy (ICP-OES, 5110, Agilent, USA) and thermogravimetric analysis (TGA, TA Q50, TA instruments, USA). X-ray absorption fine structure (EXAFS) spectra were acquired on the 1D beamline of the Pohang Accelerator Laboratory (PAL) to study the atomic coordination of the catalyst. A powder resistivity measurement system (HPRM-M2, Hantech, Korea) was used to determine the resistance of the $Ti_{xwt\%}$/HCC.

## Quasi in-situ XAFS measurement

For the in-situ measurements, a custom-designed electrochemical cell was employed. The working electrode was prepared by drop-casting the catalyst ink onto conductive graphene sheets (GRA-194, MAR-EXCEL). An Ag/AgCl electrode (LF-2, Innovative Instruments, Inc.) was employed as the reference electrode, while a platinum wire served as the counter electrode. The electrolyte, consisting of 1 M KOH, was purged with Ar and continuously circulated through the cell at a constant flow rate of 1 mL min⁻¹ using a peristaltic pump. Electrochemical control was maintained at open-circuit potential (OCP) and within the range of 1.2 $V_{RHE}$ to 1.8 $V_{RHE}$, in 0.2 V increments. Prior to each measurement, the potential was held for a minimum of 5 min to establish steady-state conditions. XAFS spectra were recorded during chronoamperometric operation.

## Electrochemical measurement

Cyclic voltammetry (CV) to study the oxygen evolution reaction (OER) and chronopotentiometry to assess the durability of LDH/HCC and LDH-Ti$_{9wt\%}$/HCC were conducted using a conventional three-electrode system with a potentiostat (PGSTAT101, Metrohm AG, Switzerland) at 25 °C. A SUS 316 mesh was used as the working electrode for durability testing, and a rotating disk electrode (RDE) was utilized for CV measurements. Hg/HgO (AMEL MOD, AMEL electrochemistry) and graphite rods (040767 Graphite rod, Alfa Aesar) were used as reference and counter electrodes, respectively. The catalyst (-1.6 mg) was loaded onto the SUS mesh, and the remaining area was affixed via epoxy resin, thus ensuring the active area was 1 cm². The potentials were converted to those of the reversible hydrogen electrode (RHE). The catalyst ink was prepared by mixing the catalyst 10 mg and 70 μL Nafion ionomer solution (5 wt.%, Sigma-Aldrich) and 700 μL of isopropyl alcohol (IPA (isopropyl alcohol, 99%, DAEJUNG, Korea)). The catalyst ink 10 μL was dropped onto the RDE surface and allowed to dry. 1 M KOH electrolyte was prepared by dissolving 62.3 g of KOH flakes (potassium hydroxide, 90%, Sigma-Aldrich) in 1 L of DI water. The solution was stored and used at RT (25 °C) in a fume hood. Additionally, a 1 M KOH + 0.5 M NaCl solution was prepared by dissolving 29.5 g of NaCl (sodium chloride, 99%, Sigma-Aldrich) in the 1 M KOH electrolyte. Before the measurements, the reference electrode was calibrated in each electrolyte by CV over a potential range from −1.3 to −0.5 $V_{Hg/HgO}$. The potential shift was determined from the x-intercept of the curve.

All of the potential reported in this study are referenced to the reversible hydrogen electrode (RHE), and the conversion was performed using the following equation:

$$E_{RHE} = E^0 + E_{Ref} + 0.0591 \times pH \tag{1}$$

CV curves were scanned in the range 1.1–1.8 $V_{RHE}$ at a scan rate of 5 mV s⁻¹ and all samples were soaked in Ar-saturated 1 M KOH (pH 14.0), 0.5 M NaCl + 1 M KOH (pH 14.0), and 1 M KOH + seawater (pH 13.8) electrolyte. The long-term durability was conducted in the same three-electrode system cell under galvanostatic mode (0.1 A cm⁻²) in a water bath in a water bath maintained at 25 °C. The EIS measurements were carried out in three-electrode system. Prior to each measurement, the electrolyte was purged with Ar gas for 1 h. The measurements were performed over a frequency range from 10 kHz to 0.01 Hz with 10 points per decade and an amplitude of 0.01 V.

The degree of carbon corrosion was evaluated using an irreversible electric charge ($Q_{ir}$) test conducted in CV mode. The $Q_{ir}$ test was conducted in the voltage range 1.1–1.9 $V_{RHE}$ in increments of 0.1 V, starting from 1.3 $V_{RHE}$. The irreversible charge was determined by measuring the difference between the first positive-direction scan and the first negative-direction scan. CV measurements were conducted over four cycles for each potential range. The Faradaic efficiencies (FEs) were determined using a rotating ring disk electrode (RRDE) in a solution of 1 M KOH saturated with Ar. The disk electrode was operated in the range 1.54–1.62 $V_{RHE}$ in increments of 0.02 V, with each potential held for 60 s using chronoamperometry to measure the current. Conversely, a potential of 0.40 $V_{RHE}$ was applied to the Pt ring electrode to consume the generated $O_2$ and determine the FE. The Faradaic efficiency is given by Eq. (1)

$$Faradaic\ efficiency = \frac{4i_r}{Ni_d} \tag{2}$$

where $i_r$ and $i_d$ are the measured ring and disk currents, respectively, and $N$ is the collection efficiency of the RRDE (0.37). The electrochemical surface area (ECSA) was determined form the double-layer capacitance ($C_{dl}$), which was obtained by measuring CV at different scan rates (20, 40, 60, 80, and 100 mV s⁻¹) in non-faradaic region (1.05–1.15 $V_{RHE}$). The charging current ($i_c$) at the center potential was obtained from the scan rate ($\nu$) and $C_{dl}$:

$$i_c = \nu C_{dl} \tag{3}$$

The turnover frequency (TOF) was calculated using the following equation:

$$TOF = (J \times A \times \zeta)/(4 \times F \times n_{mass}) \tag{4}$$

The OCl⁻ generated by the chloride evolution reaction (CER) was determined using UV-vis spectroscopy (UV-vis, Cary 3500, Agilent, USA). The samples were tested using a standard three-electrode system equipped with in-situ FT-IR (Vertex 80 v, Bruker, USA) and potentiostat (PGSTAT101, Metrohm AG, Switzerland) at room temperature (25 °C). A gold-coated silicon prism, a platinum wire, and Ag/AgCl were used as the working, counter, and reference electrodes, respectively. A continuous flow of dry air was used to maintain a stable and controlled environment throughout the tests. The measured potential range was increased from 1.1 to 1.8 $V_{RHE}$ in 0.05 V increments. At each step, chronopotentiometry was performed for 60 s.

## Single-cell tests

The AEMWE single-cell test was conducted with PtRu/C (platinum 40%, ruthenium 20%, Alfa Aesar, USA) and the synthesized LDH/HCC or LDH-Ti$_{9wt\%}$/HCC catalysts were used as cathode and anode catalysts, respectively. The membrane electrode assembly (MEA) was fabricated using a catalyst-coated membrane (CCM). The catalyst ink was prepared by mixing the catalyst with DI water, ionomer, and IPA via the bath sonicator at a temperature below 50 °C for 20 min. Poly(aryl piperidinium) (PiperION-A20-HCO3, Versogen, USA) with a thickness of 20 ± 2 μm and Poly(diphenyl-co-terphenyl piperidinium) (PDTP) membrane were used as the AEMs for 1 M KOH + 0.5 M NaCl, 1 M KOH + seawater, and 1 M KOH tests, respectively. The ionomer poly(aryl piperidinium) (PiperION-A5-HCO3-EtOH, Versogen, USA) and poly(fluorenyl-co-terphenyl piperidinium-13) (PFTP-13) and poly(fluorenyl-co-biphenyl piperidinium-14) (PFBP-14) were used as the anode and cathode for 1 M KOH + 0.5 M NaCl, 1 M KOH + seawater, and 1 M KOH tests, respectively.

For the cathode catalyst ink, PtRu/C (Y09B010, Afla Aesar) 0.025 g, DI water 0.075 g, PFBP ionomer 0.1667 g, and IPA 5 mL were mixed, and for the anode catalyst ink, LDH-Ti$_{9wt\%}$/HCC 0.02 g, DI water 0.06 g, PFTP ionomer 0.1333 g, and IPA 5 mL were mixed. Both inks were dispersed by

ultrasonicator for 30 min. The prepared catalyst inks were sprayed using a spray machine with metal loading of 0.25 mg cm$^{-2}$ on the anode and 0.5 mg cm$^{-2}$ on the cathode, respectively. The fabricated MEA have a geometric active area of 5 cm$^{-2}$. For the AEMWE test, the cell was assembled with an endplate, current collector, bipolar plate, gas diffusion layer (GDL), gaskets, and fabricated MEA. Stainless steel fiber paper (68841, Dioxide Materials, USA) and carbon paper (Toray 090, Japan) were used as the PTLs for the anode and cathode, respectively, operated in a 1 M KOH electrolyte. Titanium paper (2GDL9N-025, Bekeart, Germany) and nickel fiber paper were used as the anode and cathode PTLs, respectively, to 1 M KOH + 0.5 M NaCl conditions. For seawater electrolysis, titanium paper was used for both the anode and cathode PTLs. Prior to assembly, all the PTLs were pretreated with 30% NaOH (NaOH, 98%, Merck, USA) for 30 min and subsequently with 20% HCl (HCl, 20%, DAEJUNG, Korea) 10 min to remove impurities. The fabricated MEA was pretreated with 6 M KOH and 1 M KOH for 1 h at each step. The single-cell tests were conducted with 1 M KOH continuously supplied to both the anode and cathode at a rate of 30 mL min$^{-1}$ at cell temperatures of 60 °C and 80 °C. 1 M KOH + 0.5 M NaCl environment single-cell tests were conducted by supplying the anode at a flow rate of 30 mL min$^{-1}$, while the cathode is operated in the dry state. Seawater environment single-cell test were conducted by supplying seawater to the anode and cathode compartments at a rate of 30 mL min$^{-1}$ at cell temperatures of 80 °C. All electrochemical evaluations in the single-cell configuration were performed using an HCP-803 (Bio-Logic, France). LSV was conducted in the range of 1.35–2.00 V in 1 M KOH electrolyte and 1.35–2.30 V in 1 M KOH + 0.5 M NaCl electrolyte, respectively. The stability tests were conducted at constant current density of 1.0 A cm$^{-2}$ in 1 M KOH and 1.0 A cm$^{-2}$ in 1 M KOH + 0.5 M NaCl. Under the seawater condition, the chronopotentiometry (0.1 A cm$^{-2}$ and 0.2 A cm$^{-2}$) test was performed without any prior LSV measurements.

## Computational methods

Density functional theory (DFT) calculations were performed using the Vienna Ab initio Simulation Package (VASP)[41]. The Perdew−Burke−Ernzerhof (PBE) functional[42] was used to describe the exchange-correlation interactions. Projector-augmented wave (PAW) pseudopotentials[43] were employed to describe electron-ion interactions. A plane-wave basis set with a kinetic energy cut-off of 400 eV was employed. For basal graphene structures, a 5 × 5 graphene supercell with a 20 Å vacuum space was used to minimize periodic interactions. The Brillouin zone was sampled using a 3 × 3 × 1 Monkhorst-Pack k-point grid. All the atomic positions were fully relaxed until the residual forces were less than 0.02 eV/Å. The binding energy (BE) of the Ti-O moiety was calculated as BE = E(*TiO$_2$) − E(*) − E(TiO$_2$), where *TiO$_2$, *, and TiO$_2$ represent the TiO$_2$-decorated surfaces, base surfaces, and free TiO$_2$ moieties, respectively. The Gibbs free energy change (ΔG) of electrochemical oxidation was calculated using the computational hydrogen electrode (CHE) scheme: ΔG = ΔE + ΔZPE − TΔS + eU where ΔE is the total energy difference, ΔZPE is the zero-point energy correction, TΔS is the entropy contribution at 298.15 K, and U is the applied potential. The solvation effects were included using the VASPsol implicit solvation model[44] with a dielectric constant of 78.4 for water. The water binding energies were calculated relative to those of gas-phase H$_2$O molecules.

## Data availability

The data supporting the findings of this study are available within the article and its Supplementary Information files. Source data are provided with this paper.

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

## Acknowledgements

This work was supported by the National Research Foundation of Korea (NRF) grant funded by the Korea government (MSIT) (RS-2024-00409675 and RS-2025-02309471) and the Korea Institute of Science and Technology (KIST).

## Author contributions

Synthesis of electrocatalysts and electrochemical tests were performed by J. S. Park with support from E. Lee and S. Park. Material characterizations were performed by J. S. Park with support from S. Oh, T. K. Lee, S. Park, J.-K. Ryu, S.-H. Yu, and D. Ahn. Single-cell tests were conducted by J. S. Park, C. Hu, and H.-K. Cho. In-situ FT-IR testing was conducted by H. S. Jeon, and theoretical calculations were performed by H.-K. Lim. J. S. Park and M.-G. Kim wrote the manuscript, and Y. M. Lee and S. J. Yoo revised the draft. All authors commented on the manuscript.

## Competing interests

The authors declare no competing interest.
