## [Transparent Peer review file · Nature Communications]

Oxide-Hybridized Carbon as a Catalyst Support for Efficient Anion Exchange Membrane Water Electrolysis

Corresponding Author: Dr Sung Jong Yoo

Version 0:

Reviewer comments:

Reviewer #1

(Remarks to the Author)

In this manuscript, the authors report that Ti can passivate surface defects in highly crystalline carbon (HCC) supports, leading to enhanced corrosion resistance and improved catalytic performance of the LDH-Ti/HCC catalyst for alkaline OER. In-situ FT-IR measurements and theoretical DFT calculations reveal that oxophilic characteristics of Ti can attract H₂O and reactant OH⁻ ions, thereby improving catalytic activity. The enrichment of reactants near the catalyst surface enhances reaction efficiency and OER selectivity, enabling the LDH-Ti9wt%/HCC catalyst work well in seawater electrolysis. While the paper is generally well organized, several key claims lack sufficient supporting evidence based on the current characterizations and explanations. Thus, this manuscript cannot reach the standards for publication in Nature Communications.

1. Although this work primarily focuses on the support material (Ti/HCC), the physical and chemical characterizations of the NiFe LDH catalyst on the support, which should be the active material for alkaline OER, are insufficient. Additionally, the detailed morphology and structure of the final product (LDH-Ti/HCC) is not well characterized. The authors should provide relevant data at each synthesis step to illustrate the structure and discuss how the NiFe LDH catalyst is distributed on and interacts with the Ti/HCC support.
2. Current work emphasizes the support material and ignores the intrinsic properties and importance of the NiFe LDH catalyst. It remains unclear whether the performance enhancement comes only from the Ti/HCC support or also from other synergistic effects. Additional experiments and analysis are recommended to clarify this and rule out other contributing factors.
3. The manuscript does not clearly explain how corrosion behavior influences catalytic performance. The role of enhanced corrosion resistance in maintaining/improving catalytic stability/activity should be studied through a more in-depth and comprehensive analysis.
4. The claim that the Ti species were coordinated at the in-plane defect sites of the HCC is supported by Raman, FT-IR, and DFT calculations. However, these methods do not strongly support this claim, especially other coordination environments cannot be excluded. High-resolution TEM, XAFS, or other solid evidence should be provided.
5. The oxophilic properties of Ti enable it to attract H₂O and reactant OH⁻ ions. However, it remains unclear whether Ti also serves as active catalytic sites. This point should be clearly clarified.
6. The TEM images in Fig. S3 do not convincingly support the claim that "most Ti exists as Ti-O-C and in clusters; a small amount of TiO₂ is detected only when excess Ti precursor (above 0.5 mmol) is used."
7. In line 249, it is proposed that "Ti-O-C that is not covered by the LDHs mainly serves this role (to attracts H₂O and OH⁻ from the electrolyte)." Direct evidence should be provided to support this claim, as the exposure and role of Ti are crucial to its function in the OER process.
8. The evaluation of OER performance in the current manuscript is limited to LSV and stability tests. Additional electrochemical characterizations such as Tafel slope, electrochemical impedance spectroscopy (EIS), and turnover frequency (TOF) measurements should be included to provide a more comprehensive understanding of the catalytic mechanism and the role of the Ti-HCC support in catalytic performance.
9. A scan rate of 40 mV s⁻¹ is relatively high and might not provide reliable catalytic data. A low scan rate of 1-2 mV s⁻¹ is recommended. iR compensation information should also be provided. Additionally, the Ag/AgCl electrode is not suitable for alkaline OER testing due to stability and accuracy concerns. A Hg/HgO electrode should be considered.
10. For seawater electrolysis research, it is not very meaningful to only conduct tests in simulated seawater, because the composition of alkaline saline is too simple compared to natural seawater, and many other impurities are not taken into

account. It is recommended to use real alkaline seawater for testing.

11. a. The FTIR spectrum in Figure 1c is of low quality. Please replace it.

b. How is the electrical conductivity of the materials measured? Please provide detailed measurement methods and error values.

c. The signals shown in Fig. 4b, 4c are of low quality, which undermines their interpretability and the conclusions drawn from them.

d. In line 370, it should be cm² instead of cm⁻¹.

Reviewer #2

(Remarks to the Author)

Park and colleagues investigated the corrosion resistance mechanism of using highly crystalline carbon (HCC) as a support for anode catalysts in AEMWE. They found that introducing Ti can passivate surface defects on HCC by forming oxide-hybridized supports, effectively suppressing corrosion under high anodic potentials. The findings on using carbon as a support in AEMWE are both interesting and significant. Additionally, DFT calculations were also conducted to further elucidate the corrosion resistance mechanism. However, the conclusions are not sufficiently supported by the current evidence. Therefore, I cannot recommend acceptance of this manuscript until the authors address the following comments.

Comments on the Experimental Section:

The authors claimed that passivated Ti, forming a Ti–O–C moiety, enhances OER activity by enabling efficient OH⁻ supply near the LDH catalysts. However, they also indicated that the Ti–O–C moiety is not covered by LDH. This raises a key question: how do the OH⁻ species reach the catalyst surface?

Comments on the Computational Section:

The computational results heavily depend on the selected model and methods. Therefore, I raise the following concerns and suggestions:

It is recommended to apply the DFT+U method for treating Ti in the model. If not, validation tests should be provided to confirm that the chosen computational level reproduces consistent trends.

Why were two radical defect sites and two sp²-conjugated topological defect sites selected as representative models? Is there any experimental evidence supporting this choice?

A single Ti atom was used to passivate radical defect sites with low coordination. Given the high anodic potential and Ti's strong oxygen affinity, Ti may interact with oxygen species (e.g., O, OH), potentially forming Ti₁O_xH_y species that detach from the carbon matrix. Additional evidence is needed to support this model.

AIMD simulations are recommended to validate the thermodynamic stability of the proposed model.

Please provide structural coordinates of all intermediates to ensure reproducibility.

The authors mentioned using a 6×6 graphene supercell; however, Figure S6 appears to show a 5×5 supercell. Please clarify this discrepancy.

Reviewer #3

(Remarks to the Author)

In this paper, the Ti-hybridized carbon was synthesized to support NiFe-LDH to promote the catalytic performances of anion-exchange membrane water electrolysis (AEMWE). The synthesized LDH-Ti/HCC can achieve the current density of 8.5 A cm⁻² at 2.0 V and maintain 900 h at 1 A cm⁻² with a degradation rate of 0.17 mV h⁻¹. Therefore, the enhancement of modified LDH-Ti/HCC is significant which can be attributed to the passivation effect by introducing Ti into carbon supporter. However, there are still some problems need to be solved before the manuscript is published in the journal of Nature Communications.

1. The presence of Ti-O-C was verified by the characterizations of XPS and FT-IR spectra. While the peak of C 1s spectra located at around 285. 1 eV only can be ascribed to C-O bond and the peak of Ti 2p spectra located at around 459. 3 eV can be ascribed to Ti-O bond. Besides, the peak of Ti-O-C in FT-IR spectra is not obvious which can be disturbed by background signals. Therefore, it is hard to verify the formation of Ti-O-C. Please provide more specific characterizations to verify the presence of Ti-O-C.

2. The carbon supporter can be oxidized during the OER process accompanied with the presence of H₂O. Therefore, the hydrophobic carbon supporter is chosen to load LDH. However, the hydrophobic carbon supporter may lead to the reduction of bubble-water/catalyst tri-phase interfaces which is not beneficial to improve the OER performances. Therefore, the effective of carbon supporter used in this work is doubtful.

3. According to the results, the formation of Ti-O-C structure can lead to the shifts of C 1s spectra to higher binding energy implying the oxidation of C in pre-catalyst. Therefore, it is hard to illustrate the introduction of Ti is beneficial to alleviate the carbon corrosion.

4. The higher oxidation potential can lead to the oxidation of Ti-O-C structures which may promote the formation of TiO_x layer. And the formed TiO_x layers may prevent the corrosion of carbon supporter and facilitate the promotion of OER performances. Therefore, it is hard to distinguish whether the Ti-O-C bonding structure is beneficial to improve the OER process or not.

5. It is suggested to provide the contrast experiment with TiO₂ supported LDH to verify the role of TiO_x layers in improving the OER performances.

6. The in-suit experiment is suggested to be employed to record the variation of oxidation states of Ti during OER process to better illustrate the reaction mechanism.

Version 1:

Reviewer comments:

Reviewer #1

(Remarks to the Author)

The authors have addressed the proposed comments and the manuscript has been improved. It can be accepted. It should be noted that the durability test result for AEMWE operated under real seawater conditions (Figure 5f) shows a relatively short operation time of only 40 hours, somewhat affects the practical significance of this catalyst for seawater splitting.

Reviewer #2

(Remarks to the Author)

I appreciate the authors' efforts in addressing my concerns. All of my comments and suggestions have been adequately addressed. I am now pleased to recommend the acceptance of this manuscript for publication in this top-tier journal.

Reviewer #3

(Remarks to the Author)

In this paper, the Ti-hybridized highly crystalline carbon (HCC) supporter was employed to support FeNi-LDH to enhance the performance of anion-exchange membrane water electrolysis (AEMWE). And the corresponding assembled AEMWE can achieve 8.5 A cm^{-2} at 2.0 V which can be attributed to the improved OH⁻ supply. Besides, the formed Ti-O-C species can boost the stability of assembled AEMWE through passivating the surface defects of carbon supporter. The enhanced performances of synthesized LDH-Ti/HCC catalysts is obvious. Therefore, this paper is suggested to be accepted after the following recommended comments are revised in detail.

1. The formed Ti-O-C species is believed to passivate the surface defects to enhance the stability of synthesized catalysts. However, the introduced Ti can form a thin TiO_x layers between supported LDH and carbon supporter which is not beneficial to facilitate the electron transfer.
2. It seems that the catalysts only with TiO_x supported LDH also should be tested the performances to better analyze the real role of introduced Ti.
3. The O-H stretching band were detected through the in-situ FT-IR measurements to illustrate the promoted water supply process. It is suggested the detected O-H stretching peaks should be deconvoluted to compare the proportion of different interfacial water composition.
4. The Ti XPS spectras before and after durability tests also should be provided to better monitor the variation of introduced Ti species.
5. The introduced Ti species can enhance the operation stability of synthesized LDH-Ti/HCC through the presence of Ti-O-C species. However, the detailed mechanism about how the introduced Ti can enhance the durability of AEMWE is not clear. Please illustrate the mechanism in detailed.
6. The formation of superoxide radical is tend to attack the carbon supporter which is not beneficial to the long-term operation of AEMWE. If possible, the presence of superoxide radical also should be confirmed. And whether the introduced Ti species is helpful to remove superoxide radical or not.
7. It seems that the introduced Ti-O-C species have no significant effect in seawater electrolysis. Please explain it in detail.

Version 2:

Reviewer comments:

Reviewer #3

(Remarks to the Author)

We sincerely appreciate the editor and reviewer's valuable comments on our manuscript, which helped us improve the quality and clarity of the manuscript. Please find below our detailed point-by-point responses to each comment.

Response to Reviewers' Comments:

Reviewer: 1

Comments:

In this manuscript, the authors report that Ti can passivate surface defects in highly crystalline carbon (HCC) supports, leading to enhanced corrosion resistance and improved catalytic performance of the LDH-Ti/HCC catalyst for alkaline OER. In-situ FT-IR measurements and theoretical DFT calculations reveal that oxophilic characteristics of Ti can attract H₂O and reactant OH⁻ ions, thereby improving catalytic activity. The enrichment of reactants near the catalyst surface enhances reaction efficiency and OER selectivity, enabling the LDH-Ti_{9wt%}/HCC catalyst work well in seawater electrolysis. While the paper is generally well organized, several key claims lack sufficient supporting evidence based on the current characterizations and explanations. Thus, this manuscript cannot reach the standards for publication in Nature Communications.

Critique 1-1. Although this work primarily focuses on the support material (Ti/HCC), the physical and chemical characterizations of the NiFe LDH catalyst on the support, which should be the active material for alkaline OER, are insufficient. Additionally, the detailed morphology and structure of the final product (LDH-Ti/HCC) is not well characterized. The authors should provide relevant data at each synthesis step to illustrate the structure and discuss how the NiFe LDH catalyst is distributed on and interacts with the Ti/HCC support.

Response 1-1. We sincerely appreciate the reviewer's comment. As the reviewer suggested, we characterized the morphology and atomic structure of the final product (LDH-Ti/HCC). As evidenced by EXFAS, HR-STEM, and XPS analyses, there was no significant morphological and structural differences between LDH-Ti/HCC and LDH-HCC (**Figs. S15, S17, and S18**). Additionally, LDHs were uniformly distributed on the HCC support without aggregates. Lastly, we observed the structural evolution of LDHs at each step. As presented in **Fig. S11**, the LDH structure begins to appear at the stage III, where the reducing agent (NaBH₄) is introduced into the reaction solution. During stage IV, the LDH undergoes a growth process, resulting in a thickness of approximately 2.1 ± 1.2 nm. Additionally, in the presence of Ti induced a reduction in Ni and an oxidation in Fe. During

the synthesis process, the Ti^{4+} peak shifted toward low binding energy, indicating the presence of electronic interaction between the LDHs and Ti. We added the data and explanations to the manuscript as follows.

Please see the Page 8 and 9

“Figs. 2a shows the *ex-situ* EDS mapping at each stage of the process (Fig. S11).”

“Morphological and structural characterizations by STEM and EXAFS show that there was no significant structural and morphological difference between LDH-Ti_{9wt%}/HCC and LDH-HCC (Fig. S15 (c) and (d)); only a slight change in oxidation states of Fe and Ni was observed by XPS analyses (Figs. S17 and S18).”

Please see the Figures S15, S11, S17, and S18.

Fig. S15 | EXAFS spectra of (a) Ni K-edge and (b) Fe K-edge. HR-STEM images of (c) LDH-Ti_{9wt%}/HCC and (d) LDH-HCC, respectively.

Fig. S11 | (a) *Ex-situ* STEM images of LDH-Ti_{9wt%}/HCC obtained at each stage.

Fig. S17 | XPS spectra of (a) Ni $2p$ and (b) Fe $2p$ for LDH-Ti_{9wt%}/HCC and LDH-HCC.

Fig. S18 | XPS spectra Ti 2p at each stage

Critique 1-2. Current work emphasizes the support material and ignores the intrinsic properties and importance of the NiFe LDH catalyst. It remains unclear whether the performance enhancement comes only from the Ti/HCC support or also from other synergistic effects. Additional experiments and analysis are recommended to clarify this and rule out other contributing factors.

Response 1-2. We appreciate the reviewer's insightful comment. We agree with the comment that we should consider the intrinsic properties of the FeNi LDH catalysts. Accordingly, we measured catalytic activity of unsupported LDH catalysts, and which exhibited a higher overpotential (310 mV), compared to LDH-Ti_{9wt%}/HCC (280 mV) (**Fig. S20**). Similarly, the unsupported LDH (400 mV) showed inferior OER activity in seawater conditions, compared to that of LDH-Ti_{9wt%}/HCC (360 mV) (**Fig. R1**). Both results consistently suggest that introducing Ti/HCC improves OER catalytic activity of LDHs. We added the data and explanations to the manuscript as follows.

Please see the Page 9

“In the absence of Ti/HCC support, aggregated LDH was formed (**Fig. S19**), which shows inferior OER activity compared to LDH-Ti_{9wt%}/HCC. (**Figs. S20**).”

Please see the Methods section.

“The FeNi LDH was synthesized via the same procedure as that of LDH-HCC, except without the incorporation of HCC.”

Please see the Figs S19, S20, and R1

Fig. S19 | TEM image of unsupported FeNi LDH

Fig. S20 | LSV curve for unsupported FeNi LDH and LDH-Ti_{9wt%}/HCC at 1 M KOH

Fig. R1 | LSV curve for unsupported FeNi LDH and LDH-Ti_{9wt%}/HCC at seawater

Critique 1-3. The manuscript does not clearly explain how corrosion behavior influences catalytic performance. The role of enhanced corrosion resistance in maintaining/improving catalytic stability/activity should be studied through a more in-depth and comprehensive analysis.

Response 1-3. We sincerely appreciate the reviewer's comment. According to the reviewer's comment, we investigated the changes in morphology, chemical state, and metal composition after an electrolysis. The cross-sectional SEM images show that the thickness of MEA catalysts layer decreased after the 200 h single-cell test: LDH-HCC (from 31 to 23 μm), LDH-Ti_{9wt%}/HCC (from 34 μm to 25 μm) (**Fig. S25 (a) and (b)**). Additionally, the XPS spectra for Fe and Ni obtained from the MEA after the 200 h test show no significant peak shift between LDH-HCC and LDH-Ti_{9wt%}/HCC. Lastly, we analyzed electrolyte after cycling test to identify the dissolution of metal species during reaction. Fe and Ni in LDH-HCC tend to dissolve, whereas the metal dissolution was significantly suppressed in LDH-Ti_{9wt%}/HCC. This result suggests that the prevention of carbon corrosion could migrate the metal dissolution, leading to the long-term operation of AEMWE. The corresponding data and explanations were added to the manuscript and supporting information.

Please see the page 10

“Apart from mass transport, the effect of carbon corrosion on the AEMWE performance was investigated by using SEM, XPS, and UV-vis absorption spectroscopy. During 200 h of AEMWE test, a slight change in catalyst layer thickness and the oxidation states of Fe/Ni was observed, but the change was independent of the presence of Ti (**Fig. S25**). In contrast, the extent of metal dissolution varied significantly depending on the presence of Ti. The corrosion resistant Ti/HCC effectively suppressed metal dissolution during an electrolysis (**Fig. S26**)²⁹.”

Please see the Reference section

[29] Kadam, V. S. *et al.* One-step deposition of nanostructured Ni(OH)₂/rGO for supercapacitor applications. *Journal of Materials Science: Materials in Electronics* **34**, 1083 (2023).

Please see the Figs. S25 and S26

Fig. S25 | Cross-sectional SEM images of the MEA before and after a 200 h of operation: (a) LDH-Ti_{9wt%}/HCC and (b) LDH-HCC. (c) Fe 2p and (d) Ni 2p XPS after the durability test.

Fig. S26 | UV-vis spectra of the electrolyte collected after 1000 cycles CV in the voltage region from 1.1 to 1.8V_{RHE}.

Critique 1-4. The claim that the Ti species were coordinated at the in-plane defect sites of the HCC is supported by Raman, FT-IR, and DFT calculations. However, these methods do not strongly support this claim, especially other coordination environments cannot be excluded. High-resolution TEM, XAFS, or other solid evidence should be provided.

Response 1-4. According to the reviewer's suggestion, we measured HR-STEM image of Ti/HCC. As presented in **Fig. S7**, Ti single atoms were clearly observed in the in-plane site of the carbon support (marked with yellow circle). Consistent with FT-IR and Raman spectroscopy (**Figs. 1c and 1d**), this result support our claim that Ti exists in the in-plane sites of the carbon as Ti-O-C species. We have included the corresponding data and explanation into manuscript as below.

Please see the page 6

“In **Fig. S7**, HR-STEM images show that most Ti single atoms (marked with yellow circles) in Ti_{9wt%}/HCC are located at in-plane defect sites of the HCC.”

Please see the Fig. S7

Fig. S7 | HR-STEM images of the $\text{Ti}_{9\text{wt\%}}/\text{HCC}$. Ti single atoms are marked with yellow circles.

Critique 1-5. The oxophilic properties of Ti enable it to attract H₂O and reactant OH⁻ ions. However, it remains unclear whether Ti also serves as active catalytic sites. This point should be clearly clarified.

Response 1-5. We appreciate the reviewer's insightful comment. To exclude that Ti does not serve as active site for the OER, the current was normalized by the actual loading amount of metal species on the RDE. As shown in **Fig. S21**, the sample with a higher Ti loading showed inferior OER activity due to the relatively lower amount of Fe and Ni, suggesting that Ti does not serve as an active site. These data and discussions are added to the revised manuscript.

Additionally, our computational analysis reveals that while Ti sites exhibit strong oxophilic properties, they do not serve as active catalytic sites for the OER under typical operating condition; the Ti sites preferentially exist in the *(OH)₂ state under the experimental OER condition (~1.5 V_{RHE}). To proceed further with OER catalysis through *O_x formation at Ti site, significantly higher potentials (>2.0 V_{RHE}) would be required, which are beyond the typical operating range. This analysis confirms that the Ti sites function primarily as oxophilic anchors that enhance local OH⁻ concentration near the actual catalytic sites (FeNi LDH), rather than directly participating in the OER mechanism.

Please see the page 10

“Additionally, the current density was normalized by actual metal loading amount (**Fig. S21**). The higher Ti loading led to lower OER activity, suggesting that Ti does not contribute to OER activity.”

Please see the Fig. S21

Fig. S21 | LSV curve with Normalized current density.

Critique 1-6. The TEM images in Fig. S3 do not convincingly support the claim that “most Ti exists as Ti-O-C and in clusters; a small amount of TiO₂ is detected only when excess Ti precursor (above 0.5 mmol) is used.

Response 1-6. We sincerely appreciate the reviewer’s valuable comment. To improve the reliability of our claim that “most Ti exists as Ti-O-C and in cluster” Extended X-ray absorbance fine structure (EXAFS) was performed on Ti_{xwt%}/HCC (x= 4, 9, and 13). As presented in the EXAFS spectra, the characteristic peak around ~1.5 Å, which corresponds to Ti-O bonds in TiO₂ is barely observed, while a distinct peak appears at approximately ~1.8 Å, which is attributed to Ti-O-C bonds. This suggests that the majority of the Ti species are present in the form of Ti-O-C rather than TiO₂. We added data and brief explanations to the manuscript.

Please see the page 5

“Transmission electron microscopy (TEM) images and extended X-ray absorption fine structure (EXAFS) revealed that most Ti exists as Ti-O-C and in clusters; a small amount of TiO₂ is detected only when excess Ti precursor (above 0.5 mmol) is used (Figs. S3 and S4).”

Please see the Fig. S4

Fig. S4 | EXAFS spectra of the Ti K-edge of Ti/HCC

Critique 1-7. In line 249, it is proposed that “Ti-O-C that is not covered by the LDHs mainly serves this role (to attracts H₂O and OH⁻ from the electrolyte).” Direct evidence should be provided to support this claim, as the exposure and role of Ti are crucial to its function in the OER process.

Response 1-7. We sincerely appreciate the reviewer’s professional comment. To verify the effect of exposed Ti, we introduced KF to KOH electrolyte to poison the uncovered Ti species (*Langmuir* 2023, 39, 17853-17861). With increasing F⁻ concentration, the performance of LDH-Ti_{9wt%}/HCC was significantly degraded as a result of Ti poisoning. Moreover, the performance degradation became more pronounced in the high potential region, as Ti poisoning hindered the uncovered Ti from facilitating mass transfer H₂O and OH⁻ reactants. We added the data and explanations to the revised manuscript.

Please see the page 12

“We propose that the Ti-O-C that is not covered by the LDHs mainly serves this role.”

→ “An additional experiment was conducted to confirm the role of uncovered Ti-O-C sites in efficiently supplying reactants, by adding F⁻ ions to 1 M KOH, which are capable of poisoning Ti (**Fig. S29**)³⁴. With increasing KF concentration, the OER activity of LDH-Ti_{9wt%}/HCC was significantly degraded, especially in the high current density region, whereas that of LDH-HCC remained largely unchanged.”

Please see the Reference section

[34] Meng, X.-Z. *et al.* Molecular Insights into the Stability of Titanium in Electrolytes Containing Chlorine and Fluorine Ions. *Langmuir* **39**, 17853-17861 (2023).

Please see the Fig. S29

Fig. S29 | (a) LSV curve of LDH-HCC with increasing the KF concentration and (b) the corresponding increase in overpotential at 100 mA cm⁻². (c) LSV data of LDH-Ti_{9wt%}/HCC after 100 cycles of CV and (d) the corresponding overpotential at 100 mA cm⁻².

Critique 1-8. The evaluation of OER performance in the current manuscript is limited to LSV and stability tests. Additional electrochemical characterizations such as Tafel slope, electrochemical impedance spectroscopy (EIS), and turnover frequency (TOF) measurements should be included to provide a more comprehensive understanding of the catalytic mechanism and the role of the Ti-HCC support in catalytic performance.

Response 1-8. We thank the reviewer's valuable suggestions. According to the reviewer's recommendation, we have added the turnover frequency (TOF), Tafel slope, and EIS data to the manuscript. We also briefly explained the data in the revised manuscript.

Please see the page 10

“Tafel slopes, turnover frequency (TOF) values, and Nyquist plots were obtained (**Fig S24**). There was only a slight change in R_{ct} upon Ti introduction, implying that the electrical conductivity was not compromised.”

Please see the Fig. S24

Fig. S24 | (a) Turnover frequency (TOF) values, (b) Tafel plots, and (c) Nyquist plots.

Critique 1-9. A scan rate of 40 mV s⁻¹ is relatively high and might not provide reliable catalytic data. A low scan rate of 1-2 mV s⁻¹ is recommended. iR compensation information should also be provided. Additionally, the Ag/AgCl electrode is not suitable for alkaline OER testing due to stability and accuracy concerns. A Hg/HgO electrode should be considered

Response 1-9. As the reviewer recommended, we collected electrocatalytic data in both 1 M KOH and seawater using a lower scan rate (5 mV s⁻¹) and a Hg/HgO reference electrode. As presented in **Figs. R2 and R3**, the lower scan rate (e.g., 2 mV s⁻¹) led to oxygen bubble accumulation during an electrolysis. Additionally, we would like to note that the presented OER data was not iR-corrected. The manuscript has been updated with newly acquired data.

Please see the page 4

“A representative OER catalyst, FeNi-LDH, was loaded onto the Ti/HCC support, exhibiting exceptional durability both in half-cell (1,800 h at 0.1 A cm⁻²) and single cell (870 h at 1.0 A cm⁻²) tests.”

Please see the page 9

“The OER overpotentials for 10 mA cm⁻² for LDH-HCC, LDH-Ti_{4wt%}/HCC, LDH-Ti_{9wt%}/HCC, and LDH-Ti_{13wt%}/HCC were 282, 302, 282, and 300 mV, respectively (**Fig. 3a**).”

Please see the page 10

“**Fig. 3b** presents the results of the catalytic durability test under chronopotentiometric conditions (at 0.1 A cm⁻²). The LDH-Ti_{9wt%}/HCC demonstrated superior durability for approximately 1,800 h, whereas the LDH-HCC maintained stability for only 600 h.”

Please see the Methods section.

“A SUS 316 mesh was used as the working electrode for durability testing, and a rotating disk electrode (RDE) was utilized for CV measurements. Hg/HgO and graphite rod were used as reference and counter electrode, respectively.”

“CV curves were scanned in the range 1.1–1.8 V_{RHE} at a scan rate of 5 mV s⁻¹ and all samples were soaked in Ar-saturated 1 M KOH, 0.5 M NaCl + 1 M KOH, and Seawater + 1 M KOH electrolyte.”

Please see Fig. 3 (a), Figs. R2, and R3

Fig. 3 | Evaluation of the electrocatalytic OER performance of LDH-Ti/HCC: (a) LSV curves for OER of LDH-Ti_{xwt%}/HCC (inset: overpotential at 10 mA cm⁻²). (b) Long-term durability test via chronopotentiometry conditions for 0.1 A cm⁻² in a half-cell setup. (c) Faradaic efficiency of OER, along with the I_{disk} and I_{ring} values, measured over a voltage range of 1.56 to 1.62 V (vs. RHE). (d) Current–Voltage (I–V) curves for AEMWE measured at 60 °C and 80 °C. (e) AEMWE durability test at 1 A cm⁻². (f) Durability comparison of this work with previously reported AEMWE catalysts.

Fig. R2 | OER curves measured at various scan rate in 1 M KOH: (a) LDH-HCC, (b) LDH-Ti_{4wt%}/HCC, (c) LDH-Ti_{9wt%}/HCC, and (d) LDH-Ti_{13wt%}/HCC

Fig. R3 | OER curves measured at various scan rate in seawater: (a) LDH-HCC, (b) LDH-Ti₄wt%/HCC, (c) LDH-Ti₉wt%/HCC, and (d) LDH-Ti₁₃wt%/HCC

Critique 1-10. For seawater electrolysis research, it is not very meaningful to only conduct tests in simulated seawater, because the composition of alkaline saline is too simple compared to natural seawater, and many other impurities are not taken into account. It is recommended to use real alkaline seawater for testing.

Response 1-10. As the reviewer suggested, we conducted both half-cell and AEMWE tests using seawater mixed with 1 M KOH. A slight decrease in both AEMWE performance (**Fig. 5e**) and half-cell activity (**Fig. S30**) was observed, which can be due to the presence of various ions (e.g., SO_4^{2-} , F^- , Mg^{2+} , Ca^{2+} , etc.) in seawater. Previous studies have reported that cationic and anionic impurities can severely affect electrocatalytic performance (**Nat. Energy 2025, 01787**), (**Sustainable Energy Fuels, 2023, 7, 1565**).

Our LDH-Ti_{9wt%}/HCC exhibited stable performance 0.1 A cm⁻² for approximately 40 h. Interestingly, however, the LDH-HCC showed a rapid degradation within the first hour of operation. To elucidate the cause of this degradation, additional tests were conducted using 1 M KOH with varying concentration of Na₂SO₄. As the concentration of SO_4^{2-} increased, the OER activity significantly declined, mainly due to the poisoning of Ni active sites. This was supported by a notable decrease in peak position and intensity of the Ni oxidation peak. At low concentration (e.g., 0.03 M) of Na₂SO₄, the activity was slightly improved, likely due to the facilitated reconstruction of LDH structure (**Adv. Funct. Mater. 2021, 31, 2102772**). We added the electrochemical data obtained under seawater conditions and explained in detail the effects of SO_4^{2-} ions in the manuscript as follows.

Please see the pages 13 and 14

“Based on the high OER selectivity confirmed by the RRDE test, the seawater-splitting activities were evaluated in both simulated and real seawater. Although there was slight decrease in activity with increasing NaCl concentration of simulated seawater, the activity degradation was not influenced by the presence of Ti (**Fig. 5a**); likewise, seawater-splitting activity was independent of Ti content. This could be due to the abundance of reactants and similar electronic and structural properties of LDH-Ti_{9wt%}/HCC and LDH-HCC (**Fig. S30**).”

“Additionally, we further evaluated AEMWE performance using real seawater (**Figs. 5e and 5f**). LDH-Ti_{9wt%}/HCC exhibited a higher current density of 1.1 A cm⁻² at 2 V, compared to 0.66 A cm⁻² for LDH-HCC. It is generally accepted that AEMWE performance is degraded by cationic and anion impurities (e.g., SO_4^{2-} , F^- , Mg^{2+} , Ca^{2+} , etc.).

LDH-Ti_{9wt%}/HCC maintained stable performance at 0.1 A cm⁻² for approximately 40 h, whereas LDH-HCC showed a rapid voltage increase within the first hour of operation. To clarify the cause, we added SO₄²⁻ ions into 1 M KOH (Fig. S33). At low concentrations, the addition of SO₄²⁻ appeared to slightly enhance the performance by facilitating LDH reconstruction⁴⁰. However, as the concentration of SO₄²⁻ increases, a significant drop in performance was observed, particularly for LDH-HCC, even at lower SO₄²⁻ concentrations compared to LDH-Ti_{9wt%}/HCC. This is attributed to the blocking of Ni sites, as evidenced by the decreased intensity of Ni redox peaks. These results indicate that Ni sites in LDH-HCC are more susceptible to SO₄²⁻-induced poisoning and suggest that developing electrocatalysts resistant to anionic impurity-induced corrosion is essential for ensuring long-term durability in seawater electrolysis, although the exact mechanism by which Ti mitigates SO₄²⁻ poisoning remains unclear.”

Please see the Methods section “Seawater environment single-cell test were conducted by supplying seawater to the anode, while 1 M KOH was supplied to the cathode to minimize catalyst layer poisoning caused by Cl⁻ ions.”

Please see the Fig. S30, Fig. 5, and Fig. S33

Fig. S30 | (a) LSV curves obtained under seawater conditions

Fig. 5 | Seawater splitting performance of LDH-Ti/HCC: (a) LSV curve for OER of LDH-HCC and LDH-Ti_{9wt%}/HCC (inset: overpotential at 10 mA cm⁻²). (b) Long-term durability test via chronopotentiometry conditions for 0.1 A cm⁻² in a half-cell setup. (c) I-V curves for AEMWE operated under simulated seawater conditions (1 M KOH + 0.5 M NaCl). (d) AEMWE durability test in simulated seawater at 1 A cm⁻². (e) I-V curves of the single-cell configuration operated under seawater conditions. (f) AEMWE durability test in seawater at 0.1 A cm⁻².

Fig. S33 | LSV curves of (a) LDH-HCC with increasing SO₄²⁻ concentration and (b) the corresponding Ni oxidation region. (c) LSV curve of LDH-Ti_{9wt%}/HCC with increasing SO₄²⁻ concentration and (d) the corresponding Ni oxidation region

Please see the Reference section

[40] Liao, H. *et al.* Unveiling Role of Sulfate Ion in Nickel-Iron (oxy)Hydroxide with Enhanced Oxygen-Evolving Performance. *Advanced Functional Materials* **31**, 2102772 (2021).

Critique 1-11. The FTIR spectrum in Figure 1c is of low quality. Please replace it.

Response 1-11. According to the reviewer's comment, we have replaced the previous data with newly collected data.

Please see the Fig. 1 (c)

Fig. 1 | Fabrication and characterizations of Ti-hybridized HCC: (a) Schematic illustration of the corrosion resistance mechanism of oxide-hybridized carbon by passivating intrinsic defects in HCC. (b) XPS analysis of Ti 2p and C 1s for Ti_{xwt%}/HCC (x = 4, 9, and 13). (c) FT-IR spectra and (d) Raman analysis (left axis: I_{D1}/G ratio; right axis: I_{D2}/G ratio) results. (e) Free energy diagrams for electrochemical oxidation on radical carbon sites with and without TiO₂ passivation. (f) Irreversible charge test (Q_{ir}) result for M_{xwt%}/HCC (M = Ti, Sc, Zr, and Ta).

Critique 1-12. How is the electrical conductivity of the materials measured? Please provide detailed measurement methods and error values.

Response 1-12. We added error values to the electrical conductivity results. The conductivity was measured by placing the powder sample into a mold, which was then inserted into the instrument. During the measurement, pressure was applied to the mold while the electrical resistance was recorded. We added data and explanation in the manuscript.

Please see the page 5

Additionally, the electrical conductivity of Ti_{4wt%}/HCC, Ti_{9wt%}/HCC, and Ti_{13wt%}/HCC was comparable (**Fig. S2**).

Please see the Fig. S2 and Table. R1

Fig. S2 | Electrical conductivity of Ti_{xwt%}/HCC

Pressure (Mpa)	HCC	Ti _{4wt%} /HCC	Ti _{9wt%} /HCC	Ti _{13wt%} /HCC
2.58	4.01	4.84	4.31	3.34
5.15	5.44	6.91	6.17	4.77
7.76	6.60	8.54	7.70	5.93
10.32	7.65	9.95	9.08	6.96

Table. R1 | Electrical conductivity (S cm⁻²) of Ti_{xwt%}/HCC under different pressures (Mpa)

Critique 1-13. The signals shown in Fig. 4b, 4c are of low quality, which undermines their interpretability and the conclusions drawn from them.

Response 1-13. We agree that the quality of in-situ IR data obtained in 1 M PBS solution is relatively low. While the same experiment was conducted in 1 M KOH, the high concentration of OH⁻ ions made it difficult to clearly distinguish the change in the 3300 cm⁻¹ peak intensity between the two samples. Therefore, we presented the PBS data as the main figure and included the KOH data in the SI.

Please see the Page 11

“This result was further confirmed by *in-situ* FT-IR measurements, wherein the peak at ~3100 cm⁻¹ appeared above 1.5 V_{RHE} (inset of Fig. 4b and S27).”

Please see the Fig. S27

Fig. S27 | *In-situ* FT-IR measured from 1.2 to 1.7 V_{RHE} in 1 M KOH

Critique 1-14. In line 370, it should be cm^2 instead of cm^{-1} .

Response 1-14. We appreciate pointing out the typographical errors. We have accurately revised the manuscript as follows.

Please see the Methods section

“The catalyst (~1.6 mg) was loaded onto the SUS mesh, and the remaining area was affixed via epoxy resin, thus ensuring the active area was 1 cm^2 .”

Reviewer: 2

Comments:

Park and colleagues investigated the corrosion resistance mechanism of using highly crystalline carbon (HCC) as a support for anode catalysts in AEMWE. They found that introducing Ti can passivate surface defects on HCC by forming oxide-hybridized supports, effectively suppressing corrosion under high anodic potentials. The findings on using carbon as a support in AEMWE are both interesting and significant. Additionally, DFT calculations were also conducted to further elucidate the corrosion resistance mechanism. However, the conclusions are not sufficiently supported by the current evidence. Therefore, I cannot recommend acceptance of this manuscript until the authors address the following comments.

Critique 2-1. The authors claimed that passivated Ti, forming a Ti–O–C moiety, enhances OER activity by enabling efficient OH⁻ supply near the LDH catalysts. However, they also indicated that the Ti–O–C moiety is not covered by LDH. This raises a key question: how do the OH⁻ species reach the catalyst surface?

Response 2-1. We sincerely appreciate the reviewer's insightful comment. To clarify our claim, we conducted additional experiment using F⁻ ions, which are known to selectively poison Ti sites (*Langmuir* 2023, 39, 17853-17861) (**Fig. S29**). As the concentration of F⁻ increased, we observed a corresponding increase in overpotential, particularly for LDH-Ti_{9wt%}/HCC. Additionally, at the mass transfer region (@ 100 mA cm⁻²), a larger overpotential was observed, which can be attributed to the poisoning of uncovered Ti sites, leading to a reduced ability to attract OER reactants such as H₂O and OH⁻. These results support our claim that the uncovered Ti-O-C bond can contribute to the enhancement of OER activity in nearby LDHs. We added the corresponding data and explanation into manuscript.

Please see the page 12

“We propose that the Ti-O-C that is not covered by the LDHs mainly serves this role.”
→ “An additional experiment was conducted to confirm the role of uncovered Ti-O-C sites in efficiently supplying reactants, by adding F⁻ ions to 1M KOH, which are capable of poisoning Ti (**Fig. S29**). With increasing KF concentration, the OER activity of LDH-Ti_{9wt%}/HCC was significantly degraded, especially in the high current density region, whereas that of LDH-HCC remained largely unchanged.”

Please see the Fig. S29

Fig. S29 | (a) LSV curve of LDH-HCC with increasing the KF concentration and (b) the corresponding increase in overpotential at 100 mA cm⁻². (c) LSV data of LDH-Ti_{9wt%}/HCC after 100 cycles of CV and (d) the corresponding overpotential at 100 mA cm⁻².

Critique 2-2. The computational results heavily depend on the selected model and methods. Therefore, I raise the following concerns and suggestions: It is recommended to apply the DFT+U method for treating Ti in the model. If not, validation tests should be provided to confirm that the chosen computational level reproduces consistent trends.

Response 2-2. We appreciate the reviewer's valuable suggestion regarding the computational methodology. Following the reviewer's recommendation, we have recalculated all DFT results using the DFT+U method, which is indeed more appropriate for treating Ti-O systems due to the partially localized *d*-electrons of Ti. We applied the DFT+U approach with $U_{\text{eff}} = 2$ eV for Ti 3d orbitals, which is consistent with literature values for TiO₂ systems (J. Phys. Chem. C 2011, 115, 13, 5841-5845). While the resulting energy levels show minor differences compared to the standard PBE calculations, the overall trends and conclusions remain consistent: a) The preferential binding of Ti-O moieties at radical defect sites (V_1, V_3) over *sp*² conjugated sites (V_{2-1}, V_{2-2}) is maintained, b) The enhanced oxidation resistance of Ti-passivated sites compared to pristine radical carbon sites remains evident, c) The oxophilic behavior and water binding characteristics show the same trends. All computational results presented in the revised manuscript (**Figs 1e, 4d, 4e and S9**) have been updated using the DFT+U methodology, ensuring more accurate description of the electronic structure and energetics of the Ti-containing systems. We added the corresponding data and explanation into manuscript.

Please see the page 6

“Notably, the V_1 and V_3 sites exhibited strong BEs of -4.23 and -4.49 eV, respectively, which are energetically favorable, comparable to the cohesive energy of bulk TiO₂ (-5.86 eV). This suggests that the Ti-O moiety can effectively passivate unstable radical carbon sites, thereby providing comprehensive surface stabilization. In contrast, the BEs of V_{2-1} and V_{2-2} were relatively weak (-0.95 and -1.45 eV, respectively), due to the energy cost associated with breaking the *sp*² conjugation network.”

Please see the Figs. 1e, 4d, 4e, and S9

Fig. 1 | Fabrication and characterizations of Ti-hybridized HCC: (a) Schematic illustration of the corrosion resistance mechanism of oxide-hybridized carbon by passivating intrinsic defects in HCC. (b) XPS analysis of Ti 2p and C 1s for Ti_{xwt%}/HCC (x= 4, 9, and 13). (c) FT-IR spectra and (d) Raman analysis (left axis: I_{D1}/G ratio; right axis: I_{D2}/G ratio) results. (e) Free energy diagrams for electrochemical oxidation on radical carbon sites with and without TiO₂ passivation. (f) Irreversible charge test (Q_{ir}) result for M_{xwt%}/HCC (M = Ti, Sc, Zr, and Ta).

Fig. 4 | Elucidating the role of Ti in water electrolysis: (a) Schematic illustration of the attraction of OH⁻ from the solution to LDH by the HCC surface decorated with Ti. FT-IR spectra of (b) LDH-Ti_{9wt%}/HCC and (c) LDH/HCC, obtained before (dotted) and after (solid) electrolysis in 1 M PBS (inset: *in-situ* FT-IR measured from OCV to 1.8 V in 1 M PBS). (d) Free energy diagrams showing stepwise oxidation pathways of Ti sites at V₁ and V₃ defects. (e) Water binding energies at oxygen-functionalized graphene (Gr) and Ti-passivated sites (V₁/V₃-

Fig. S9 | (a) Binding energy calculations for TiO₂ moiety at (b) different defect configurations of HCC, demonstrating preferential binding at radical defect sites

Critique 2-3. Why were two radical defect sites and two sp^2 -conjugated topological defect sites selected as representative models? Is there any experimental evidence supporting this choice?

Response 2-3. The selection of these specific defect configurations is based on well-established understanding of carbon defects commonly formed during high-temperature synthesis processes (RSC Adv. 2022, 12, 21520-21547, Appl. Surf. Sci. 2021, 536, 147851, and J. Appl. Phys. 2011, 110, 093524). The radical defect sites (V_1 , V_3) represent single and triple carbon vacancy sites, which are characterized by unpaired electrons and high reactivity, making them prime candidates for heteroatom decoration. The sp^2 conjugated topological defects (V_{2-1} , V_{2-2}) represent Stone-Wales-type defects and divacancy reconstructions that maintain sp^2 hybridization while creating topological irregularities in the graphene lattice, which provide perturbed electronic structures more adaptable to guest chemical moieties. These defect types are commonly observed in carbon materials and represent the most likely sites for Ti incorporation based on coordination chemistry principles and binding energy considerations.

Critique 2-4. A single Ti atom was used to passivate radical defect sites with low coordination. Given the high anodic potential and Ti's strong oxygen affinity, Ti may interact with oxygen species (e.g., O, OH), potentially forming Ti_1O_xHy species that detach from the carbon matrix. Additional evidence is needed to support this model.

Response 2-4. We have addressed this concern through comprehensive computational analysis of the Ti sites under operating conditions. Our DFT calculations confirm that Ti sites indeed function as oxophilic centers and can accommodate various oxygen-containing species depending on the applied potential. The theoretical Pourbaix diagram (**Fig. 4e**) demonstrates that: Under OER conditions ($\sim 1.5 V_{RHE}$), the Ti sites preferentially form $*(OH)_2$ species. At higher oxidation potentials, further oxidation can lead up to $*O_3$ formation, representing highly oxidized Ti centers. Crucially, even under these highly oxidizing conditions, the Ti-O-C bonds through the radical carbon sites remain strong (binding energy of $TiO_2-(OH)_2$ moiety is -5.46 eV and -5.37 eV for the V_1 and V_3 radical sites, respectively), preventing detachment from the carbon matrix. The key finding is that the covalent Ti-O-C bonding through the radical carbon defects provides sufficient anchoring strength to maintain structural integrity even when the Ti centers are decorated with multiple oxygen species. This explains the enhanced durability observed experimentally - the Ti sites can function as stable oxophilic centers without compromising the overall structural stability of the support. The manuscript has been updated

Please see the page 12

“ V_1/V_3 - $TiO_2-(OH)_2$ demonstrated enhanced water-binding affinity, regardless of the hydrogen-bonding configuration (Fig. 4e**).”**

Critique 2-5. AIMD simulations are recommended to validate the thermodynamic stability of the proposed model.

Response 2-5. Following the reviewer's suggestion, we have performed ab initio molecular dynamics (AIMD) simulations to validate the thermodynamic stability of our base Ti-passivated radical defect site models (V_1/V_3 -TiO₂). We conducted 100 ps AIMD simulations at 300 K, and the results confirm the thermodynamic stability with no significant structural deformation or detachment of TiO₂ moieties observed during the simulation period (**Fig. R4**). These results provide strong evidence that the proposed Ti-passivated defect sites are thermodynamically stable under operating conditions and support the experimental observations of enhanced long-term durability.

Fig. R4 | AIMD simulation results showing temperature and energy evolution for Ti-passivated radical defect sites (V_1 -TiO₂ and V_3 -TiO₂) at 300 K over 100 ps.

Critique 2-6. Please provide structural coordinates of all intermediates to ensure reproducibility.

Response 2-6. We have included the optimized structural coordinates (in VASP POSCAR format) of all key structures in the Supporting Information (Section S16). This includes bare graphene structures with their corresponding Ti-passivated structures, and electrochemical intermediates ($V_1/V_3-(OH)_xO_y$ models). These coordinates enable full reproducibility of our computational results and facilitate further research in this area.

Critique 2-7. The authors mentioned using a 6×6 graphene supercell; however, Figure S6 appears to show a 5×5 supercell. Please clarify this discrepancy.

Response 2-7. We thank the reviewer for catching this error. The computational models indeed used a 5×5 graphene supercell, not 6×6 as incorrectly stated in the computational methods section. We have corrected this in the manuscript.

Please see the page 20

“For basal graphene structures, a 5 × 5 graphene supercell with a 20 Å vacuum space was used to minimize periodic interactions.”

Reviewer 3:

Comments:

In this paper, the Ti-hybridized carbon was synthesized to support NiFe-LDH to promote the catalytic performances of anion-exchange membrane water electrolysis (AEMWE). The synthesized LDH-Ti/HCC can achieve the current density of 8.5 A cm^{-2} at 2.0 V and maintain 900 h at 1 A cm^{-2} with a degradation rate of 0.17 mV h^{-1} . Therefore, the enhancement of modified LDH-Ti/HCC is significant which can be attributed to the passivation effect by introducing Ti into carbon supporter. However, there are still some problems need to be solved before the manuscript is published in the journal of Nature Communications.

Critique 3-1. The presence of Ti-O-C was verified by the characterizations of XPS and FT-IR spectra. While the peak of C 1s spectra located at around 285.1 eV only can be ascribed to C-O bond and the peak of Ti 2p spectra located at around 459.3 eV can be ascribed to Ti-O bond. Besides, the peak of Ti-O-C in FT-IR spectra is not obvious which can be disturbed by background signals. Therefore, it is hard to verify the formation of Ti-O-C. Please provide more specific characterizations to verify the presence of Ti-O-C.

Response 3-1. We appreciate reviewer's valuable comment. We agree that XPS analysis is insufficient to confirm that the C 1s and Ti 2p peaks were derived from Ti-O-C bond. Accordingly, we conducted EXAFS measurements to confirm our suggestions. As shown in EXAFS spectra, the characteristic Ti-O bond ($\sim 1.5 \text{ \AA}$) corresponding to TiO_2 were barely observed in $\text{Ti}_{\text{xwt\%}}/\text{HCC}$ sample, that the incorporated Ti species primarily exist as Ti-O-C rather than TiO_2 . Additionally, we replaced FT-IR data with newly measured one. We added the corresponding data and explanation to manuscript.

Please see the page 5

“Transmission electron microscopy (TEM) images and extended X-ray absorption fine structure (EXAFS) revealed that most Ti exists as Ti-O-C and in clusters; a small amount of TiO_2 is detected only when excess Ti precursor (above 0.5 mmol) is used (Figs. S3 and S4).”

Please see the Fig. S4

Fig. S4 | EXAFS spectra of the Ti *K*-edge of Ti/HCC

Critique 3-2. The carbon supporter can be oxidized during the OER process accompanied with the presence of H₂O. Therefore, the hydrophobic carbon supporter is chosen to load LDH. However, the hydrophobic carbon supporter may lead to the reduction of bubble-water/catalyst tri-phase interfaces which is not beneficial to improve the OER performances. Therefore, the effective of carbon supporter used in this work is doubtful.

Response 3-2. We sincerely appreciate the reviewer's valuable comment. As the review pointed out, surface engineering strategies have been widely employed to modify catalysts more hydrophilic to improve interfacial contact with the electrolyte, because the hydrophobic surface of catalysts could reduce bubble/water/catalyst tri-phase. However, beyond catalysts, surface properties of supporting materials can also influence the catalysts performances. For example, the hydrophilic characteristics of supports can sometimes impair performances by trapping reactants (ChemCatChem 2024, 16, e202301308). Moreover, in the case of carbon-based supports, they are susceptible to corrosion as a result of interaction with water ($C + 2H_2O \rightarrow CO_2 + 4H^+$), making hydrophilic carbons even more vulnerable. Our previous study demonstrated that hydrophobic carbon supports can suppress such interactions with water, effectively mitigating corrosion. Also, they can facilitate the transport of reactants to hydrophilic catalysts (e.g., LDHs), leading to improved electrochemical performances. Therefore, the use of hydrophobic carbon supports could be an effective strategy to enhance OER activity without sacrificing durability.

Please see the Fig. R5

Fig. R5 | Correlation between the hydrophobicity of carbon support and AEMWE performance. (a) Total, water, and oxygen diffusion coefficients of graphene (black) and graphene with oxygen functional groups (dark yellow). (b) Radial distribution function of water (left) and oxygen (right). (c) Photograph of the behavior of water and oxygen during molecular dynamics simulations. Reproduced from ref (10) Copyright 2024, Energy Environ. Sci.

Critique 3-3. According to the results, the formation of Ti-O-C structure can lead to the shifts of C1s spectra to higher binding energy implying the oxidization of C in pre-catalyst. Therefore, it is hard to illustrate the introduction of Ti is beneficial to alleviate the carbon corrosion.

Response 3-3. Thank you for reviewer's comment. As presented in **Fig. S1**, the binding energy of the C 1s peak corresponding to the Ti-O-C bond slightly shifts to lower values with increasing Ti contents: 285.4, 285.2, and 285.1 eV for Ti_{4wt%}/HCC, Ti_{9wt%}/HCC, and Ti_{13wt%}/HCC, respectively. Meanwhile, the peak associated with the C-C bond remains unchanged at 284.8 eV. These findings and Qir test results consistently show that Ti may contribute to suppressing oxidation of carbon support.

Please see the Fig. S1 presented below

Fig. S1 | XPS (a) Ti 2p and (b) C 1s spectra for $Ti_{xwt\%}/HCC$

Critique 3-4. The higher oxidation potential can lead to the oxidation of Ti-O-C structures which may promote the formation of TiO_x layer. And the formed TiO_x layers may prevent the corrosion of carbon supporter and facilitate the promotion of OER performances. Therefore, it is hard to distinguish whether the Ti-O-C bonding structure is beneficial to improve the OER process or not.

Response 3-4. We sincerely appreciate the reviewer's insightful comment. In accordance with the reviewer's suggestions, we conducted *in-situ* XAFS measurements to verify whether Ti changes to TiO_x layer during the OER process. The *in-situ* XAFS analysis shows that most Ti remains unchanged, showing negligible variation in oxidation state across the potential range of 1.2 to 1.8 V_{RHE}.

Furthermore, the contribution of the Ti-O-C structure to the OER activity was evaluated by selectively poisoning surface Ti-O-C bonds. When surface Ti-O-C was blocked by fluoride (F⁻) ions, the current density of LDH-Ti_{9wt%}/HCC—particularly in the mass transport region—significantly decreased (**Fig. S29**). These findings support our claim that Ti plays a role in attracting OER reactants, such as H₂O and OH⁻, thereby enhancing catalytic performance.

Please see the page 12

“To further confirm that the Ti species remain unchanged during the OER process, *in-situ* XAFS analysis was performed. As shown in **Fig. S28**, the spectra remain unchanged during electrolysis. An additional experiment was conducted to confirm the role of uncovered Ti-O-C sites in efficiently supplying reactants, by adding F⁻ ions to 1 M KOH, which are capable of poisoning Ti (**Fig. S29**)³⁴. With increasing KF concentration, the OER activity of LDH-Ti_{9wt%}/HCC was significantly degraded, especially in the high current density region, whereas that of LDH-HCC remained largely unchanged.”

Please see the Methods section

Quasi *in-situ* XAFS measurement

For the *in-situ* measurements, a custom-designed electrochemical cell was employed. The working electrode was prepared by drop-casting the catalyst ink onto conductive graphene sheets (GRA-194, MAREXCEL). An Ag/AgCl electrode (LF-2, Innovative Instruments, Inc.) was employed as the reference electrode, while a platinum wire served as the counter electrode. The electrolyte, consisting of 1 M KOH, was purged with Ar and continuously circulated through the cell at a constant flow rate of 1 mL min⁻¹ using a peristaltic pump. Electrochemical control was maintained at open-circuit potential (OCP) and within the range of 1.2 V_{RHE} to 1.8 V_{RHE}, in 0.2 V increments. Prior to each measurement, the potential was held at least 5 minutes to establish steady-state conditions, after which the electrolyte was removed using the peristaltic pump to minimize X-ray attenuation and ensure accurate spectral acquisition.

Please see the Figs. S28 and S29

Fig. S28 | *In-situ* Ti K-edge XAFS spectra of LDH-Ti_{9wt%}/HCC measured at different potentials: 1.2, 1.4, 1.6, and 1.8 V_{RHE}

Fig. S29 | (a) LSV curve of LDH-HCC with increasing the KF concentration and (b) the corresponding increase in overpotential at 100 mA cm^{-2} . (c) LSV data of LDH-Ti_{9wt%}/HCC after 100 cycles of CV and (d) the corresponding overpotential at 100 mA cm^{-2} .

Critique 3-5. It is suggested to provide the contrast experiment with TiO₂ supported LDH to verify the role of TiOx layers in improving the OER performances.

Response 3-5. According to the reviewer's suggestion, for preparation of LDH-TiO₂/HCC, the synthesis procedure was identical to that of LDH-Ti_{9wt%}/HCC, except that the Ti precursor was replaced with TiCl₄ instead of TTIP. According to the ICP-OES analysis, the Ti content was determined to be 5.43 wt.%. XRD analysis confirmed the successful formation of LDH-TiO₂/HCC. The metal (Fe and Ni) content on TiO₂/HCC was 12.1 wt.%, approximately half of that in LDH-Ti/HCC. For a fair comparison, the OER activity was evaluated by using the same amount of metal loading. As presented in **Fig. R**, the LDHs supported on TiO₂ exhibited inferior performance to that of LDH-Ti/HCC, which could be due to the lower electrical conductivity of TiO₂ than Ti-O-C.

Please see the Fig. R6

Fig. R6 | (a) XRD pattern and (b) LSV curve of LDH-TiO₂/HCC

Critique 3-6. The in-suit experiment is suggested to be employed to record the variation of oxidation states of Ti during OER process to better illustrate the reaction mechanism.

Response 3-6. According to the reviewer's suggestion, we conducted the *in-situ* XAFS measurement. Please see the *response 3-4*.

Response to Reviewers' Comments

Reviewer: 1

Critique 1-1. It should be noted that the durability test result for AEMWE operated under real seawater conditions (Figure 5f) shows a relatively short operation time of only 40 hours, somewhat affects the practical significance of this catalyst for seawater splitting.

Response 1-1. In response to the reviewer's suggestion, we have provided further clarification regarding the unsatisfactory performances under seawater conditions. From the first to the second round of revision, we have sought to identify the cause of the significant durability loss in line with the reviewer's critique. While we confirmed that the decrease in activity was mainly due to catalyst poisoning by F^- and SO_4^{2-} anions, the origin of the pronounced durability degradation remained unclear. In this revision, however, we found that the AEMWE durability was strongly affected by the condition of seawater and evaluation sequence. To ensure reliable testing, we optimized the preparation process for the 1 M KOH + seawater solution as well as the seawater AEMWE procedure. Regarding the latter, in our earlier tests, chronopotentiometry (CP) measurements were conducted after multiple LSV scans (from 1.3 to 2.2 V). Since several reports indicate that repeated potential cycling imposes severe stress on the catalyst layer, we performed CP measurements without prior electrochemical tests (*ACS Energy Lett.*, **2025**, 10, 6, 2574-2581). With this modified protocol, the sample showed stable operation for approximately 250 h under seawater conditions. We have replaced the previous dataset with these newly obtained results and described the evaluation sequence in detail in the Supporting Information. Nevertheless, we agree with the reviewer that our synthesized catalyst may face limitations in practical applications due to the presence of various ions and organic compounds in seawater, and we have addressed these concerns in the revised manuscript as follows.

Please see the Page 14 "LDH-Ti_{9wt%}/HCC maintained stability for about 250 h, whereas LDH-HCC maintained stable performance for 80 h"

Please see the Method section "Under the seawater condition, the chronopotentiometry (0.1 A cm⁻² and 0.2 A cm⁻²) test was performed without any prior LSV measurements."

Reviewer 3:

Comments:

In this paper, the Ti-hybridized highly crystalline carbon (HCC) supporter was employed to support FeNi-LDH to enhance the performance of anion-exchange membrane water electrolysis (AEMWE). And the corresponding assembled AEMWE can achieve 8.5 A cm^{-2} at 2.0 V which can be attributed to the improved OH⁻ supply. Besides, the formed Ti-O-C species can boost the stability of assembled AEMWE through passivating the surface defects of carbon supporter. The enhanced performances of synthesized LDH-Ti/HCC catalysts is obvious. Therefore, this paper is suggested to be accepted after the following recommended comments are revised in detail.

Critique 3-1. The formed Ti-O-C species is believed to passivate the surface defects to enhance the stability of synthesized catalysts. However, the introduced Ti can form a thin TiO_x layers between supported LDH and carbon supporter which is not beneficial to facilitate the electron transfer.

Response 3-1. As the reviewer pointed out, we were also concerned that TiO_x layers might form due to the strong tendency of Ti to oxidize. Interestingly, however, most Ti in Ti_{xwt%}/HCC exists in the form of Ti-O-C, as evidenced by EXAFS and XRD analyses. In these measurements, the characteristic peaks of TiO_x were barely observed (**Fig. S4 and S5**). Consistently, electrical conductivity results (e.g., powder resistance and EIS) show negligible variation with Ti loading, indicating that Ti does not exist in an insulating TiO_x form in Ti_{xwt%}/HCC (**Fig. S2 and S24(c)**). These findings suggest that the formation of TiO_x during the Ti-O-C bonding process is minimal, and even if present, it does not significantly affect electrochemical performance. As shown in **Fig. 2c**, LDH-Ti_{xwt%}/HCC (mainly composed of Ti-O-C) is obtained after pre-formed TiO_x species are etched by Cl⁻ ions. Thus, we suggest that TiO_x could be unlikely to re-nucleate in the presence of etching agents. As the reviewer noted, the presence of TiO_x between LDH and HCC could be not favorable for electron transfer, which is further demonstrated in **Response 3-2**.

Please see the Figs. S4, S5, S2, and S24 (c)

Fig. S4 | EXAFS spectra of the Ti K-edge of Ti/HCC

Fig. S5 | XRD pattern of Ti_{xwt%}/HCC. The black and pink bar represent reference data for TiO₂ and carbon, respectively

Fig. S2 | Electrical conductivity of Ti_{xwt%}/HCC

Fig. S24 | (a) TOF values, (b) Tafel plots, and (c) Nyquist plots

Critique 3-2. It seems that the catalysts only with TiO_x supported LDH also should be tested the performances to better analyze the real role of introduced Ti

Response 3-2. We sincerely appreciate the reviewer's concern that the role of Ti should be verified using various Ti-containing supports (e.g., TiO_x, TiO_x-HCC, and Ti-HCC). In the previous revision, we demonstrated that LDH-TiO₂/HCC exhibited comparable activity to LDH-Ti/HCC. In this revision, we additionally prepared LDH supported on TiO₂ (denoted as LDH-TiO₂). This sample showed poor activity due to the low electrical conductivity of TiO₂, as evidenced by EIS analysis (**Fig. S31**). It should be noted that the Fe and Ni loadings were comparable to those of LDH-Ti_{9wt%}/HCC (Fe: 3.47 wt.% and Ni: 22.9 wt.%). To mitigate the conductivity issue, HCC was added to the catalyst ink (denoted as LDH-TiO₂ + HCC). This modification improved the electrocatalytic performance; however, the activity remained inferior to that of LDH-TiO₂/HCC and LDH-HCC + TiO₂; this difference could be attributed to the interfacial resistance between LDH and HCC. In summary, as the reviewer noted in *Critique 3-2*, we demonstrate that both Ti and TiO₂ can play a role in electrolysis. However, when TiO₂ becomes excessively thick (in case of LDH-TiO₂ + HCC), it hinders electron transfer between LDH and HCC, leading to degraded OER performance. We sincerely thank the reviewer again for this insightful comment, and we have added the corresponding data and explanations to the revised manuscript

Please see the page 12 “These results suggest that Ti species on HCC play a key role in attracting OH⁻ ions, thereby contributing to the improved single-cell performance. Additionally, the LDH supported on TiO₂/HCC and the LDH/HCC physically mixed with TiO₂ exhibited comparable activity to LDH-Ti_{9wt%}/HCC, further confirming the role of Ti. In contrast, the LDH-TiO₂ mixed with HCC showed poor activity, indicating that a thin Ti layer is essential to ensure efficient electron transfer between LDH and HCC (**Figs. S30 and S31**).

Please see the Figs. S30 and S31

Fig. S30 | LSV curves of LDHs supported on different supporting materials: solid symbols represent LDH on carbon-based supports, while hollow symbols represent LDH on TiO₂.

Fig. S31 | Nyquist plots of LDH-TiO₂ and LDH-Ti₉wt%/HCC.

Critique 3-3. The O-H stretching band were detected through the in-situ FT-IR measurements to illustrate the promoted water supply process. It is suggested the detected O-H stretching peaks should be deconvoluted to compare the proportion of different interfacial water composition

Response 3-3. We sincerely appreciate the reviewer's comment. Following the reviewer's suggestion, we deconvoluted the *in-situ* FT-IR spectra measured in 1 M KOH (**Fig. S29**). The peak at $\sim 3250\text{ cm}^{-1}$ corresponds to tetrahedral coordination, while the peak at $\sim 3400\text{ cm}^{-1}$ is attributed to trihedral (incomplete tetrahedral) coordination. The LDH-Ti_{9wt%}/HCC sample exhibits a markedly higher proportion of trihedral coordination than LDH/HCC over the measured potential region, and the ratio of tetrahedral to trihedral coordination remains nearly unchanged with increasing applied potential. Recent studies have demonstrated that O-H stretching from trihedral coordination is more favorable for electrochemical reactions compared to tetrahedral coordination (*EES Catal.*, **2025**, DOI: 10.1039/D5EY00161G). Nevertheless, the correlation between interfacial water coordination and electrochemical performance remains an area that requires further in-depth investigation. We have therefore included these data and a brief explanation in the revised manuscript.

Please see the page 12 “The peak at $\sim 3250\text{ cm}^{-1}$ can be deconvoluted to tetrahedral coordination, while the peak at $\sim 3400\text{ cm}^{-1}$ is attributed to trihedral coordination (**Fig. S29**). A higher proportion of tetrahedrally coordinated water was observed on LDH-Ti/HCC. However, the detailed correlation between interfacial water coordination and electrochemical performance is beyond the scope of this study.”

Please see the Fig. S29

Fig. S29 | Deconvoluted *in-situ* FT-IR spectra for (a) LDH-Ti_{9wt%}/HCC and (b) LDH-HCC.

Critique 3-4. The Ti XPS spectras before and after durability tests also should be provided to better monitor the variation of introduced Ti species.

Response 3-4. As suggested by the reviewer's we conducted XPS measurements of the Ti 2p before and after the 200 h AEMWE durability test. As shown in **Fig. S25**, negligible changes were observed in the Ti⁴⁺ spectra, indicating that the chemical state of Ti remained largely unchanged after the durability test. In addition, a Ru 3p_{3/2} peak appeared at approximately 462 eV after the durability test, which likely originated from the PtRu/C cathode catalyst. We added the data with brief explanations to the revised manuscript.

Please see the page 11 "Additionally, Ti 2p XPS spectra confirmed that the chemical state of Ti remained largely unchanged after the 200 h durability test (**Fig. S25 (d)**)."

Please see the Fig. S25

Fig. S25 | Cross-sectional SEM images of the MEA before and after a 200 h of operation: (a) LDH-Ti_{9wt%}/HCC and (b) LDH-HCC. (c) C 1s, (d) Ti 2p, (e) Ni 2p, and (f) Fe 2p XPS after the durability test

Critique 3-5. The introduced Ti species can enhance the operation stability of synthesized LDH-Ti/HCC through the presence of Ti-O-C species. However, the detailed mechanism about how the introduced Ti can enhance the durability of AEMWE is not clear. Please illustrate the mechanism in detailed.

Response 3-5 We appreciate the reviewer's comments. First, Ti is preferentially incorporated into the in-plane defect sites of HCC, thereby enhancing the corrosion resistance of the carbon support. Second, the corrosion-resistant carbon suppresses metal dissolution during electrolysis. As a result, less Fe and Ni from LDHs dissolved when supported on Ti/HCC compared to bare HCC, as evidenced by XPS and UV-vis absorption spectra. We described the plausible degradation mechanism in the manuscript as follows.

Please see the page 11 “Although no noticeable change in catalyst layer thickness was observed during the 200 h of AEMWE test (**Figs. S25 (a) and (b)**), XPS C 1s spectra revealed a stronger carbonate peak in LDH/HCC, compared to LDH-Ti/HCC (**Fig. S25 (c)**). Additionally, Ti 2p XPS spectra confirmed that the chemical state of Ti remained largely unchanged after the durability test. (**Fig. S25 (d)**). Meanwhile, the extent of metal dissolution varied significantly depending on the presence of Ti. The corrosion-resistant Ti/HCC effectively suppressed metal dissolution during electrolysis (**Figs. S25 (e), (f), and S26**)²⁹. The difference likely originated from corrosion resistance, as evidenced by the changes in the I_D/I_G ratio (**Fig. S27**).”

Critique 3-6. The formation of superoxide radical is tend to attack the carbon supporter which is not beneficial to the long-term operation of AEMWE. If possible, the presence of superoxide radical also should be confirmed. And whether the introduced Ti species is helpful to remove superoxide radical or not.

Response 3-6. As the reviewer pointed out, we found that oxygen species radical (such as hydroxyl radical and/or superoxide radical) are generated during the OER. To confirm the effect of radicals on carbon corrosion, we conducted electrochemical test under chronopotentiometry conditions at 0.1 A cm^{-2} in 1 M KOH containing methyl orange (MO); herein, the amount of generated radicals can be indirectly quantified by monitoring the absorption intensity of MO, which reacts with the radicals. During the reaction, electrolyte samples were taken every hour and analyzed by UV-vis spectroscopy. As shown in **Fig. S27** the final amounts of radical were nearly identical. Nevertheless, as evidenced by Raman analysis, the extent of carbon degradation differed between the two samples (**Figs. S27 and R1**). For LDH–HCC, the D/G ratio increased by approximately 35.8%, whereas LDH–Ti_{9wt%}/HCC exhibited only an 18.7% increase. This result indicates that Ti effectively passivates the defect sites of carbon thereby suppressing carbon corrosion. We added the data and explanations to the manuscript as follows.

Please see the page 11 “The difference likely originated from corrosion resistance, as evidenced by the changes in the I_{D1}/I_G ratio (**Fig. S27**).”

Please see the Figs. S27 and R1

Fig. S27 | UV-vis spectra of the electrolyte containing methyl orange solution: (a) LDH-HCC and (b) LDH-Ti_{9wt%}/HCC. (c) The D₁/G ratio before and after 5 h of reaction

Fig. R1 | Raman spectra before and after 5 h of electrolysis: (a) LDH-HCC and (b) LDH-Ti_{9wt%}/HCC

Critique 3-7. It seems that the introduced Ti-O-C species have no significant effect in seawater electrolysis. Please explain it in detail.

Response 3-7. As the reviewer pointed out, the effect of Ti–O–C species was negligible in seawater electrolysis compared to simulated seawater electrolysis. We attempted to identify the reason for this large performance gap. First, we confirmed that the metals are prone to poisoning by specific anions: F⁻ tends to poison Ti, while SO₄²⁻ tends to poison Ni (**Fig. S33**). Although the precise origin of the severe durability loss was not initially clear, our further investigation revealed that the AEMWE durability was strongly affected by the condition of seawater and evaluation sequence. To ensure reliable testing, we optimized the preparation process for the 1 M KOH + seawater solution as well as the seawater AEMWE procedure. Regarding the latter, in our earlier tests, chronopotentiometry (CP) measurements were conducted after multiple LSV scans (from 1.3 to 2.2 V). Since several reports indicate that repeated potential cycling imposes severe stress on the catalyst layer, we performed CP measurements without prior electrochemical tests (*ACS Energy Lett.*, **2025**, 10, 6, 2574-2581). With this modified protocol, the sample showed stable operation for approximately 250 h under seawater conditions. We have replaced the previous dataset with these newly obtained results and described the evaluation sequence in detail in the Supporting Information. Importantly, we also acknowledge the reviewer’s concern that, despite this improvement, the catalyst could still face limitations under practical seawater conditions due to the coexistence of various ions and organic compounds. These points have been carefully addressed in the revised manuscript.

Please see Figs. 5f, S33, and S37

Fig. 5 | Seawater splitting performance of LDH-Ti/HCC: (a) LSV curve for OER of LDH-HCC and LDH-Ti_{xwt%}/HCC (inset: overpotential at 10 mA cm⁻²). (b) Long-term durability test via chronopotentiometry conditions for 0.1 A cm⁻² in a half-cell setup. (c) I–V curves for AEMWE operated under simulated seawater conditions (1 M KOH + 0.5 M NaCl). (d) AEMWE durability test in simulated seawater at 1 A cm⁻². (e) I–V curves and (f) durability test result for AEMWE operated under real seawater conditions.

Fig. S33 | (a) LSV curve with different concentration of KF in 1M KOH and the corresponding increase in overpotential at 100 mA cm⁻²: (a, b) LDH-Ti_{9wt%}/HCC, (c, d) LDH-HCC. LSV curves were measured after 100 CV cycles in the range of 1.1 to 1.8 V_{RHE}, allowing sufficient time for F⁻ ions to poison the catalyst surface

Fig. S37 | LSV curves obtained at different concentration of SO₄²⁻ in 1M KOH, showing changes in the peak area corresponding to Ni oxidation: (a, b) LDH-HCC and (c, d) LDH-Ti_{9wt%}/HCC. LSV curves were measured after 15 CV cycles in the range of 1.1 to 1.8 V_{RHE}, allowing sufficient time for SO₄²⁻ ions to poison the catalyst surface